# Balance Act: Mitigating Hubness in Cross-Modal Retrieval with Query and Gallery Banks

**Yimu Wang**[*] and **Xiangru Jian**
University of Waterloo
{yimu.wang,xiangru.jian}@uwaterloo.ca

**Bo Xue**
City University of Hong Kong
boxue4-c@my.cityu.edu.hk

## Abstract

In this work, we present a post-processing solution to address the hubness problem in cross-modal retrieval, a phenomenon where a small number of gallery data points are frequently retrieved, resulting in a decline in retrieval performance. We first theoretically demonstrate the necessity of incorporating both the gallery and query data for addressing hubness as hubs always exhibit high similarity with gallery and query data. Second, building on our theoretical results, we propose a novel framework, Dual Bank Normalization (DBNORM). While previous work has attempted to alleviate hubness by only utilizing the query samples, DB-NORM leverages two banks constructed from the query and gallery samples to reduce the occurrence of hubs during inference. Next, to complement DBNORM, we introduce two novel methods, dual inverted softmax and dual dynamic inverted softmax, for normalizing similarity based on the two banks. Specifically, our proposed methods reduce the similarity between hubs and queries while improving the similarity between non-hubs and queries. Finally, we present extensive experimental results on diverse language-grounded benchmarks, including text-image, text-video, and text-audio, demonstrating the superior performance of our approaches compared to previous methods in addressing hubness and boosting retrieval performance. Our code is available at https://github.com/yimuwangcs/Better_Cross_Modal_Retrieval.

## 1 Introduction

Cross-modal retrieval (CMR) facilitates flexible information retrieval across various modalities, including images, videos, audio, and text, by extracting discriminative features and summarizing information from multiple modalities. Recently, supported by the development of pre-training multimodal models (Tsai et al., 2019; Li et al., 2020a;

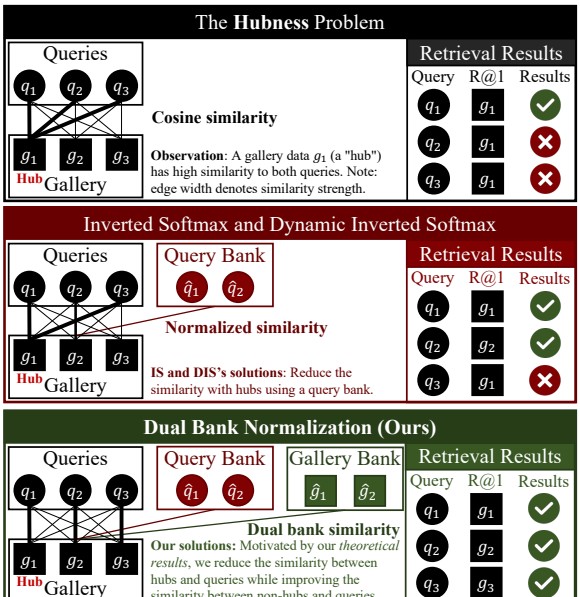

Figure 1: **Top**: The hubness problem (Radovanovic et al., 2010) in cross-modal retrieval where queries $\mathbf{q}_1$, $\mathbf{q}_2$, and $\mathbf{q}_3$ are associated with their respective galleries $\mathbf{g}_1$, $\mathbf{g}_2$, and $\mathbf{g}_3$. The hub ($\mathbf{g}_1$) is the nearest neighbor to multiple queries ($\mathbf{q}_1$, $\mathbf{q}_2$, and $\mathbf{q}_3$), resulting to suboptimal retrieval performance. **Middle**: Previous methods employ a query bank to normalize similarities. **Bottom**: Our DBNORM tackles the hubness problem by utilizing both gallery and query banks. It reduces the similarity between the hub $\mathbf{g}_1$ and all queries while simultaneously enhancing the similarity between non-hubs $\mathbf{g}_2$ and $\mathbf{g}_3$ and all queries, resulting in improved performance.

Yu et al., 2021; Frank et al., 2021; Fang et al., 2023b), innovative retrieval methods (Wang et al., 2020b,a; Luo et al., 2022b; Jian and Wang, 2023), and cross-modal benchmarks (Xu et al., 2016; Kim et al., 2023), significant advances have been made in image-text retrieval (Liu and Ye, 2019; Radford et al., 2021; Wang et al., 2021; Luo et al., 2022b), video-text retrieval (Xu et al., 2021; Wu et al., 2022; Park et al., 2022a; Chen et al., 2022; Park et al., 2022a; Zhao et al., 2022; Fang et al., 2022, 2023a; Wang and Shi, 2023), and audio-text

---
[*]Corresponding author.

retrieval (Koepke et al., 2022), achieving satisfactory retrieval performance.

In the most popular CMR paradigm, deep neural networks are employed to project data from diverse modalities into a shared high-dimensional vector space, enabling direct comparison through distance measures such as cosine similarity and $\ell_2$ similarity. However, within this paradigm, the occurrence of hubness (Radovanovic et al., 2010) poses a challenge. Hubness refers to the phenomenon where certain data points, known as hubs, frequently emerge as nearest neighbors of other points, resulting in a decline in retrieval performance. To empirically illustrate the existence of hubness, we visualize the distribution of the frequency at which each gallery data point is retrieved by queries, as shown in Figure 3 and Figure 6 in the Appendix. We observe that hubness is prevalent in video-text, image-text, and audio-text retrieval, significantly impacting the performance of retrieval systems (Bogolin et al., 2022).

Prior research on mitigating hubness can be roughly categorized into training methods (Liu et al., 2020) and post-processing methods (Suzuki et al., 2013; Dinu et al., 2015; Smith et al., 2017; Lample et al., 2018; Huang et al., 2019; Liu and Ye, 2019), with particular attention in zero-shot learning (Lazaridou et al., 2015; Huang et al., 2020) and bilingual word translation (Smith et al., 2017; Huang et al., 2019, 2020). Due to the limitation of space, detailed related works are presented in Appendix A. In this paper, we focus on addressing hubness in a post-processing manner.

However, hubness in the context of CMR received limited attention until the introduction of QBNorm (Bogolin et al., 2022), which is a pioneering method specifically designed for CMR and incorporates a novel technique called DIS. It addresses hubness by selectively reducing the similarity between queries and hubs, thereby mitigating the impact of hubs. It follows the principle that hubs exhibit high similarity with query data to identify hubs. This prompts us to investigate whether hubs also demonstrate high similarity with gallery data. Therefore, the first research question is:

### RQ1: Do hubs exhibit high similarity with gallery data?

To answer it and provide a theoretical validation for the principle of QBNorm, we theoretically demonstrate that hubness is a universal characteristic across different modalities. In other words, a hub exhibits high similarity with data from various modalities, which shows the necessity of using gallery data. The second research question is:

### RQ2: How can we leverage gallery data to mitigate hubness?

Motivated by our theoretical results, instead of only using query data as QBNorm, we propose a unified framework, namely Dual Bank Normalization (DBNORM) as shown in Figure 1, along with two novel normalizing methods, namely dual inverted softmax and dual dynamic inverted softmax, which leverage both query and gallery data banks to alleviate the occurrence of hubs by reducing the similarity between hubs and queries and improving the similarity between non-hubs and queries.

Finally, to evaluate the effectiveness of our proposed DBNORM, we conducted experiments on several cross-modal benchmarks, including four video-text retrieval benchmarks (Chen and Dolan, 2011; Fabian Caba Heilbron and Niebles, 2015; Xu et al., 2016; Hendricks et al., 2017), two image-text retrieval benchmarks (Lin et al., 2014; Plummer et al., 2017), and two audio-text retrieval benchmarks (Kim et al., 2019; Drossos et al., 2020). Benefiting from gallery and query banks, DBNORM outperforms previous methods without requiring access to any additional data.

In summary, our contributions are as follows[1]:

- We propose a unified framework DBNORM and two novel post-processing methods, namely dual inverted softmax and dual dynamic inverted softmax. These methods effectively mitigate hubness by reducing the similarity between hubs and queries with query and gallery banks.

- Our proposed methods achieve state-of-the-art performance on eight cross-modal retrieval benchmarks, outperforming previous approaches across multiple evaluation metrics.

- We are the first to theoretically demonstrate that hubs exhibit high similarity with data from different modalities and the necessity of utilizing both query and gallery data to address the issue of hubness.

---

[1]The code is released at link.

## 2 Our Methods

In this section, we first address RQ1 through a theoretical demonstration of the universality of hubness and the necessity of incorporating both query and gallery data to address hubness. Next, to answer RQ2, we propose a novel method called DBNORM, which builds upon our theoretical analysis, along with two novel similarity normalization techniques: Dual Inverted Softmax and Dynamic Dual Inverted Softmax. These methods effectively utilize both query and gallery data to mitigate hubness.

### 2.1 Preliminaries

In this paper, we focus on cross-modal retrieval (CMR), aiming to learn a pair of encoders that map data from different modalities into a common space where they can be directly compared.

The query and gallery modalities are denoted as $\mathcal{X}$ and $\mathcal{Y}$. The (test) gallery, denoted by $G = \{\mathbf{g}_1, \ldots, \mathbf{g}_{N_G}\}$, contains all the embeddings of the gallery data, where $N_G$ is the size of the gallery data. In cross-modal retrieval, the gallery data does not overlap with the training data. Moreover, as our proposed methods require training data to address hubness, we define the sets of embeddings of training query and gallery data as $\hat{Q} = \{\hat{\mathbf{q}}_1, \ldots, \hat{\mathbf{q}}_{N_{\hat{Q}}}\}$ and $\hat{G} = \{\hat{\mathbf{g}}_1, \ldots, \hat{\mathbf{g}}_{N_{\hat{G}}}\}$, where $N_{\hat{Q}}$ and $N_{\hat{G}}$ are the sizes. Finally, the query data is denoted as $\mathbf{q}$. Following QBNorm, we focus on a practical scenario where only a single query occurs at a time and queries are unable to observe each other. In such a situation, we are limited to utilizing the training data for constructing banks.

### 2.2 RQ1: Theoretical Analysis

In cross-modal retrieval, we consider two spaces, denoted as $\mathcal{X}$ and $\mathcal{Y}$, that contain embeddings or representations of data points from two modalities. We assume that the two spaces, $\mathcal{X}$ and $\mathcal{Y}$, follow symmetric distributions[2] although they may have different means, variances, and distribution types. Specifically, the mean of $\mathcal{X}$ and $\mathcal{Y}$ are denoted as $\boldsymbol{\mu}_x$ and $\boldsymbol{\mu}_y$.

First, we demonstrate that points (embeddings or representations) close to the mean point are more likely to exhibit hubness compared to points situated farther away from the mean. We use the $\ell_2$ distance measure to compute distances.

---

[2]This assumption is reasonable since various symmetric distributions, such as Uniform, Normal, or Laplacian distributions, exist.

**Theorem 1.** *Assuming that $\mathbf{x}_1$ and $\mathbf{x}_2$ are sampled from a distribution $\mathcal{X}$ with mean $\boldsymbol{\mu}$, if $\mathbf{x}_1$ is closer to $\boldsymbol{\mu}$ than $\mathbf{x}_2$, then $\mathbf{x}_1$ is more likely to be a hub than $\mathbf{x}_2$ on the space $\mathcal{X}$, such that,*

$$\mathbb{E}\left[\|\mathbf{x}_2 - \mathbf{x}\|^2\right] > \mathbb{E}\left[\|\mathbf{x}_1 - \mathbf{x}\|^2\right], \forall \mathbf{x} \sim \mathcal{X}.$$

Next, we demonstrate that a hub will have high similarity (small distance) with points from another space (distribution).

**Theorem 2.** *Assuming $\mathbf{x}_1, \mathbf{x}_2 \sim \mathcal{X}$ and $\mathbf{y} \sim \mathcal{Y}$, similarly, if $\mathbf{x}_1$ is closer to the mean point $\boldsymbol{\mu}_x$ of the space $\mathcal{X}$ than $\mathbf{x}_2$, such that $\|\mathbf{x}_2 - \boldsymbol{\mu}_x\| > \|\mathbf{x}_1 - \boldsymbol{\mu}_x\|$, then even in a different space (distribution) $\mathcal{Y}$, $\mathbf{x}_1$ is still more likely to be a hub than $\mathbf{x}_2$, such that:*

$$\mathbb{E}\left[\|\mathbf{x}_2 - \mathbf{y}\|^2\right] > \mathbb{E}\left[\|\mathbf{x}_1 - \mathbf{y}\|^2\right], \forall \mathbf{y} \sim \mathcal{Y}.$$

Based on the previous theoretical results, we introduce the following corollary.

**Corollary 3.** *Under the $\ell_2$ distance measure, a hub will exhibit high similarity (small distance) with any point from $\mathcal{X}$ or $\mathcal{Y}$.*

**Remark 1.** *This corollary implies that a hub will be frequently retrieved by both the query and gallery points, as the similarity between that point and any other point will be high (small distance).*

Additionally, we extend our theoretical findings beyond the $\ell_2$ distance measure and explore the use of hyper-spheres, which can be regarded as using cosine similarity for measuring the distance between data points in CMR. This is particularly relevant since the cosine similarity is widely used in recent work (Luo et al., 2022a; Gao et al., 2021).

**Theorem 4.** *Hubs will have high similarity with any point from $\mathcal{X}$ and $\mathcal{Y}$ under the cosine distance measure.*

Now, we have shown that it is necessary to use gallery and query data to identify whether a point is a hub, as the hub will exhibit high similarity with any point under $\ell_2$ and cosine metrics. Furthermore, our theoretical results also suggest that hubness can be better mitigated by utilizing larger data banks[3], as demonstrated by Smith et al. (2017); Bogolin et al. (2022). All the proofs are deferred to Appendix C due to the limitation of space.

---

[3]With more data, the estimation of the mean will be more precise.

**Algorithm 1** DBNORM

**Require:** the query point $\mathbf{q}$, the gallery set $G$.
1: Construct dual banks $\hat{Q}$ and $\hat{G}$ from the training or validation sets. ▷ **Dual Bank Construction.**
2: Calculate the unnormalized similarity $s_{\mathbf{q},\mathbf{g}_i} = sim(\mathbf{q}, \mathbf{g}_i), \forall i \in [N_G]$. ▷ **Similarity Computation.**
3: Calculate normalized similarity $\hat{s}_{\mathbf{q},\mathbf{g}_i}, \forall i \in [N_G]$, with DualIS (Equation (1)) or DualDIS (Equation (2)). ▷ **Similarity Normalization.**
4: **return** Ranking $\text{argsort}_{i \in [N_G]}\, \hat{s}_{\mathbf{q},\mathbf{g}_i}$.

## 2.3 RQ2: DBNORM

To mitigate the pervasive hubness problem observed in various cross-modal retrieval benchmarks and tasks (Figures 3 and 6), we propose a novel approach named Dual Bank Normalization (DBNORM), as summarized in Algorithm 1. DBNORM first constructs banks from the training data of both modalities and then normalizes the similarity. Specifically, DBNORM reduces the similarity between the query and hubs in the gallery data, *i.e.*, data points that appear as nearest neighbors of many queries, and increases the similarity between the query and non-hub gallery data. DBNORM includes the following steps.

**Dual Bank Construction.** In order to identify hubs in the gallery, DBNORM constructs two banks, $\hat{Q} = \{\hat{\mathbf{q}}_1, \ldots, \hat{\mathbf{q}}_{N_{\hat{Q}}}\}$ and $\hat{G} = \{\hat{\mathbf{g}}_1, \ldots, \hat{\mathbf{g}}_{N_{\hat{G}}}\}$, where $N_{\hat{Q}}$ and $N_{\hat{G}}$ are the sizes of the two banks. As suggested by our theoretical results, using both query and gallery banks simultaneously allows for a better estimation of the hubness of a data point.

**Similarity Computation.** The similarity between the query and the gallery is calculated as $s_{\mathbf{q},\mathbf{g}_i} = sim(\mathbf{q}, \mathbf{g}_i), \forall i \in [N_G]$, where $sim(\cdot, \cdot)$ is a similarity metric.

**Similarity Normalization.** To normalize the similarity, we introduce two novel methods: DualIS (Equation (1)) and DualDIS (Equation (2)). These methods allow for the normalization of the similarity $\hat{s}_{\mathbf{q},\mathbf{g}_i}, \forall i \in [N_G]$.

Finally, with the normalized similarity $\hat{s}_{\mathbf{q},\mathbf{g}_i}, \forall i \in [N_G]$, we can obtain the ranking of gallery data. It is worth noting that in practice, the query and gallery banks should be precomputed and reused across all queries, resulting in significant improvements in computational efficiency.

### 2.3.1 Normalization Methods

As existing methods are either designed for only using the query bank or cannot be directly incorporated into DBNORM[4], we propose two novel methods to efficiently implement DBNORM, as outlined below.

First, drawing inspiration from IS (Smith et al., 2017), by inverting the softmax, and normalizing the probability over query and gallery banks rather than the test gallery, we propose **Dual I**nverted **S**oftmax (**DualIS**). Given the query $\mathbf{q}$ and a gallery $\mathbf{g}_i$, the normalized similarity $\hat{s}_{\mathbf{q},\mathbf{g}_i}$ is calculated as follows,

$$
\begin{aligned}
\hat{s}_{\mathbf{q},\mathbf{g}_i} &= \hat{s}^{\mathbf{g}}_{\mathbf{q},\mathbf{g}_i} * \hat{s}^{\mathbf{q}}_{\mathbf{q},\mathbf{g}_i} \\
\hat{s}^{\mathbf{g}}_{\mathbf{q},\mathbf{g}_i} &= \frac{\exp(\beta_1 s_{\mathbf{q},\mathbf{g}_i})}{\sum_{j \in [N_{\hat{G}}]} \exp(\beta_1 s_{\mathbf{g}_i, \hat{\mathbf{g}}_j})} \\
\hat{s}^{\mathbf{q}}_{\mathbf{q},\mathbf{g}_i} &= \frac{\exp(\beta_2 s_{\mathbf{q},\mathbf{g}_i})}{\sum_{j \in [N_{\hat{Q}}]} \exp(\beta_2 s_{\mathbf{g}_i, \hat{\mathbf{q}}_j})},
\end{aligned}
\tag{1}
$$

where $\beta_1$ and $\beta_2$ are temperatures and $s_{\mathbf{q},\mathbf{g}_i}$ is the similarity between $\mathbf{q}$ and $\mathbf{g}_i$. We employ multiplication as an aggregation method to combine the normalized similarities from two banks because it effectively summarizes the information from both sides. A detailed comparison with other aggregation methods is presented in Appendix B.5. The intuition behind DualIS is to measure the probability that the corresponding gallery retrieves the query data, rather than testing whether the query retrieves the corresponding gallery.

Previous research (Bogolin et al., 2022) has shown that when the bank fails to effectively represent the space, possibly due to sampling bias, the performance is significantly degraded, even falling below that of unnormalized similarities, because the similarities with non-hubs might be inaccurately normalized by IS and DualIS. To address this issue, we introduce the **Dual D**ynamic **I**nverted **S**oftmax (**DualDIS**). Firstly, we precompute two activation sets, $\mathcal{A}_{\hat{G}}$ and $\mathcal{A}_{\hat{Q}}$, storing the indices of gallery data that might be hubs. Then, DualDIS only normalize the similarity with the gallery points in these sets to avoid inaccurately normalizing the similarity with non-hubs. Specifically, for a query

---

[4]In practice, queries are isolated from each other, making it infeasible to include CSLS (Lample et al., 2018) and GC (Dinu et al., 2015) as they require queries to be visible to each other. A detailed comparison with CSLS and GC is in Appendix B.6.

$\mathbf{q}, \forall i \in [N_G]$, we have,

$$\hat{s}_{\mathbf{q},\mathbf{g}_i} = \bar{s}^{\mathbf{g}}_{\mathbf{q},\mathbf{g}_i} * \bar{s}^{\mathbf{q}}_{\mathbf{q},\mathbf{g}_i} ,$$

$$\bar{s}^{\mathbf{g}}_{\mathbf{q},\mathbf{g}_i} = \begin{cases} \hat{s}^{\mathbf{g}}_{\mathbf{q},\mathbf{g}_i}, & \text{if } \arg\max_{j \in [N_G]} s_{\mathbf{q},\mathbf{g}_j} \in \mathcal{A}_{\mathbf{g}} \\ s_{\mathbf{q},\mathbf{g}_i}, & \text{otherwise} \end{cases} ,$$

$$\bar{s}^{\mathbf{q}}_{\mathbf{q},\mathbf{g}_i} = \begin{cases} \hat{s}^{\mathbf{g}}_{\mathbf{q},\mathbf{g}_i}, & \text{if } \arg\max_{j \in [N_G]} s_{\mathbf{q},\mathbf{g}_j} \in \mathcal{A}_{\mathbf{q}} \\ s_{\mathbf{q},\mathbf{g}_i}, & \text{otherwise} \end{cases} ,$$
(2)

where $\mathcal{A}_* = \{\arg\max^k_{j \in [N_G]} s_{\mathbf{g}_j, \hat{*}_i}, i \in [N_*]\}$, and $\arg\max^k_{j \in [N_G]} s_{\mathbf{g}_j, \hat{*}_i}$ return the top $k$ indices that maximize $s_{\mathbf{g}_j, \hat{*}_i}$ ($* \in \{\mathbf{g}, \mathbf{q}\}$). In other words, $\mathcal{A}_*$ contains the indices of the top $k$ retrieved gallery points by any bank point. As the activation sets can be computed before inference and reused, thereby not increasing the computational complexity of inference. On the other hand, benefiting from these two activation sets, DualDIS is shown to be more robust than DualIS.

## 3 Experiments

In this section, we first conduct experiments to demonstrate that DBNORM can efficiently improve the retrieval performance as a by-product. Next, our ablation studies indicate that DBNORM effectively mitigates hubness by reducing skewness and exhibiting robustness to hyperparameters. For clarity, we use IS, DIS, DualIS, and DualDIS to represent QBNorm (IS), QBNorm (DIS), DBNorm (DualIS), and DBNorm (DualDIS), respectively. Following QBNorm, queries remain independent of each other as in practice, queries do not always occur at the same time. Detailed comparison with GC and CSLS is deferred to Appendix B.6.

### 3.1 Datasets, Experimental Settings, and Evaluation Metrics

While we mainly focus our experiments on standard benchmarks for text-video retrieval, *i.e.*, MSR-VTT (Xu et al., 2016), MSVD (Chen and Dolan, 2011), ActivityNet (Fabian Caba Heilbron and Niebles, 2015), and DiDemo (Hendricks et al., 2017), we also explore the generalization of DB-NORM on two text-image retrieval benchmarks (MSCOCO (Lin et al., 2014) and Flickr30k (Plummer et al., 2017)) and two audio-text retrieval benchmarks (AudioCaps (Kim et al., 2019) and CLOTHO (Drossos et al., 2020)). We test DB-NORM with DualIS and DualDIS on five video-text retrieval methods: CE+ (Liu et al., 2019),

| Methods | Normalization | R@1↑ | R@5↑ | R@10↑ | MdR↓ | MnR↓ |
|---|---|---|---|---|---|---|
| MSR-VTT (full split) | | | | | | |
| RoME | | 10.7 | 29.6 | 41.2 | 17.0 | - |
| Frozen | | 32.5 | 61.5 | 71.2 | - | - |
| CE+ | | 12.62 | 33.87 | 46.38 | 12.0 | 74.99 |
| | + IS | 13.31 | 35.00 | 47.46 | 12.0 | 74.16 |
| | + DIS | 13.31 | 35.00 | 47.46 | 12.0 | 74.16 |
| | + DualIS | **14.88** | **37.36** | **50.00** | 11.0 | **70.13** |
| | + DualDIS | **14.88** | **37.36** | **50.00** | 11.0 | **70.13** |
| TT-CE+ | | 14.61 | 37.81 | 50.78 | 10.0 | 63.31 |
| | + IS | 16.58 | 40.75 | 53.44 | 9.0 | 60.52 |
| | + DIS | 16.59 | 40.73 | 53.44 | 9.0 | 60.51 |
| | + DualIS | **17.06** | **41.63** | **54.25** | 8.0 | 60.10 |
| | + DualDIS | 17.02 | 41.60 | 54.23 | 8.0 | **60.01** |
| MSR-VTT (1k split) | | | | | | |
| DiscreteCodebook | | 43.4 | 72.3 | 81.2 | - | 14.8 |
| VCM | | 43.8 | 71.0 | - | 2.0 | 14.3 |
| CenterCLIP | | 44.2 | 71.6 | 82.1 | 2.0 | 15.1 |
| Align&Tell | | 45.2 | 73.0 | 82.9 | 2.0 | - |
| CLIP2TV | | 45.6 | 71.1 | 80.8 | 2.0 | 15.0 |
| X-Pool | | 46.9 | 72.8 | 82.2 | 2.0 | 14.3 |
| TS2-Net | | 47.0 | 74.5 | 83.8 | 2.0 | 13.0 |
| CAMoE | | 47.3 | 74.2 | 84.5 | 2.0 | 11.9 |
| CLIP4Clip | | 44.10 | 71.70 | 81.40 | 2.0 | 15.51 |
| | + IS | 44.20 | 71.70 | 81.60 | 2.0 | 15.64 |
| | + DIS | 44.20 | 71.70 | 81.60 | 2.0 | 15.64 |
| | + DualIS | **45.00** | **72.50** | **82.10** | 2.0 | **15.32** |
| | + DualDIS | **45.00** | **72.50** | **82.10** | 2.0 | **15.32** |
| CLIP2Video | | 46.00 | 71.60 | 81.60 | 2.0 | 14.51 |
| | + IS | 47.00 | 72.80 | 82.10 | 2.0 | 13.91 |
| | + DIS | 47.00 | 72.80 | 82.10 | 2.0 | 13.91 |
| | + DualIS | **47.20** | **73.20** | **82.30** | 2.0 | **13.90** |
| | + DualDIS | **47.20** | 72.70 | **82.30** | 2.0 | **13.90** |
| X-CLIP | | 46.30 | 74.00 | 83.40 | 2.0 | **12.80** |
| | + IS | 48.60 | 74.10 | 84.10 | 2.0 | 13.35 |
| | + DIS | 48.60 | 74.10 | 84.10 | 2.0 | 13.35 |
| | + DualIS | **48.80** | **74.30** | 84.10 | 2.0 | 13.30 |
| | + DualDIS | 48.70 | **74.30** | 84.10 | 2.0 | 13.30 |

Table 1: Text-to-Video Retrieval performance on MSR-VTT (full split and 1k split). Best in **Bold** and the second best is underlined.

TT-CE+ (Croitoru et al., 2021), CLIP4Clip (Luo et al., 2022a), X-CLIP (Ma et al., 2022), and CLIP2Video (Park et al., 2022a), two image-text retrieval methods: CLIP (Radford et al., 2021) and Oscar (Li et al., 2020b), and one audio-text retrieval method: AR-CE (Koepke et al., 2022).

For evaluation metrics, we employ recall at Rank K (R@K, higher is better), median rank (MdR, lower is better), and mean rank (MnR, lower is better) as commonly utilized retrieval metrics in previous retrieval works (Radford et al., 2021; Luo et al., 2022a; Ma et al., 2022).

### 3.2 Quantitative results

In this section, we provide quantitative results for eight cross-modal retrieval benchmarks. We compare our methods with IS (Smith et al., 2017) and DIS (Bogolin et al., 2022), which are two representative post-processing methods. Due to limitations of space, some retrieval results and detailed analyses are presented in Appendix B.3.

**Text-video retrieval**. The text-to-video results

| Methods | Normalization | R@1↑ | R@5↑ | R@10↑ | MdR↓ | MnR↓ |
|---|---|---|---|---|---|---|
| FSE | | 11.5 | 31.8 | 77.7 | 13.0 | - |
| HiT | | 27.7 | 58.6 | 94.7 | 4.0 | - |
| VCM | | 40.8 | 72.8 | 98.2 | 2.0 | 7.3 |
| CenterCLIP | | 43.9 | 74.6 | 85.8 | 2.0 | 6.7 |
| Align&Tell | | 42.6 | 73.8 | 98.7 | 2.0 | - |
| CLIP2TV | | 44.1 | 75.2 | 98.4 | 2.0 | 6.5 |
| TS2-Net | | 41.0 | 73.6 | 84.5 | 2.0 | 8.4 |
| CAMoE | | 51.0 | 77.7 | - | - | - |
| CE+ | | 19.16 | 47.79 | 65.79 | 6.0 | 21.99 |
| | + IS | 20.13 | 50.58 | 66.44 | 5.0 | 21.07 |
| | + DIS | 20.20 | 50.50 | 66.32 | 5.0 | 21.20 |
| | + DualIS | **22.66** | **52.21** | **68.09** | 5.0 | **19.02** |
| | + DualDIS | 22.35 | 51.96 | 68.25 | 5.0 | 19.22 |
| TT-CE+ | | 23.29 | 56.42 | 73.78 | 4.0 | 13.59 |
| | + IS | 24.69 | 57.98 | 74.42 | 4.0 | 13.48 |
| | + DIS | 24.65 | 57.64 | 74.31 | 4.0 | 13.35 |
| | + DualIS | **27.15** | **60.81** | **76.67** | 4.0 | **12.10** |
| | + DualDIS | 26.80 | 60.16 | 76.35 | 4.0 | 12.18 |
| CLIP4Clip | | 41.85 | 74.44 | 84.84 | 2.0 | 6.84 |
| | + IS | 45.93 | 77.52 | 87.07 | 2.0 | 6.39 |
| | + DIS | 46.02 | 77.29 | 86.83 | 2.0 | 6.29 |
| | + DualIS | 46.71 | 77.47 | 87.35 | 2.0 | 6.01 |
| | + DualIS | **46.76** | **77.48** | 87.28 | 2.0 | 6.01 |
| X-CLIP | | 46.25 | 76.02 | 86.05 | 2.0 | 6.37 |
| | + IS | 49.36 | **79.16** | 88.36 | 2.0 | 5.71 |
| | + DIS | 49.39 | 78.60 | 87.97 | 2.0 | 5.80 |
| | + DualIS | 49.96 | 78.51 | 88.62 | 2.0 | **5.48** |
| | + DualDIS | **50.17** | 78.03 | 88.06 | 1.0 | 5.61 |

Table 2: Text-to-Video Retrieval performance on ActivityNet. Best in **Bold** and the second best is underlined.

| Methods | Normalization | R@1↑ | R@5↑ | R@10↑ | MdR↓ | MnR↓ |
|---|---|---|---|---|---|---|
| FSE | | 18.2 | 44.8 | 89.1 | 7.0 | - |
| Frozen | | 33.7 | 64.7 | 76.3 | 3.0 | - |
| CenterCLIP | | 47.3 | 76.9 | 86.0 | 2.0 | 9.7 |
| Align&Tell | | 47.1 | 77.0 | 85.6 | 2.0 | - |
| CLIP2TV | | 47.0 | 76.5 | 85.1 | 2.0 | 10.1 |
| X-Pool | | 47.2 | 77.4 | 86.0 | 2.0 | 9.3 |
| CAMoE | | 49.8 | 79.2 | 87.0 | - | 9.4 |
| CE+ | | 23.94 | 54.98 | 69.00 | 4.0 | 18.46 |
| | + IS | 24.64 | **56.64** | 70.45 | 4.0 | 20.43 |
| | + DIS | 24.66 | **56.64** | 70.46 | 4.0 | 20.42 |
| | + DualIS | 25.04 | 56.01 | 69.59 | 4.0 | 20.70 |
| | + DualDIS | 25.06 | 56.02 | 69.65 | 4.0 | 20.64 |
| TT-CE+ | | 24.42 | 56.20 | 70.44 | 4.0 | 17.16 |
| | + IS | 26.55 | 59.68 | 72.85 | 4.0 | 17.72 |
| | + DIS | 26.57 | 59.68 | 72.83 | 4.0 | 17.71 |
| | + DualIS | 27.21 | **60.23** | 73.35 | 4.0 | **16.88** |
| | + DualDIS | 27.24 | 60.20 | 73.35 | 4.0 | 16.91 |
| CLIP4Clip | | 44.64 | 74.66 | 83.99 | 2.0 | 10.32 |
| | + IS | 46.05 | 75.60 | 84.36 | 2.0 | 10.16 |
| | + DIS | 46.05 | 75.60 | 84.37 | 2.0 | 10.16 |
| | + DualIS | 46.33 | 75.91 | 84.35 | 2.0 | 10.14 |
| | + DualDIS | **46.34** | **75.95** | 84.34 | 2.0 | **10.12** |
| CLIP2Video | | 47.05 | 76.97 | 85.59 | 2.0 | 9.53 |
| | + IS | 47.52 | **77.95** | 86.01 | 2.0 | 9.47 |
| | + DIS | 47.52 | **77.95** | 86.00 | 2.0 | 9.47 |
| | + DualIS | 47.95 | **77.95** | 86.22 | 2.0 | 9.30 |
| | + DualDIS | **47.97** | 77.93 | 86.20 | 2.0 | 9.30 |
| X-CLIP | | 46.31 | 76.84 | 85.31 | 2.0 | 9.59 |
| | + IS | 47.06 | 77.44 | 85.22 | 2.0 | 10.34 |
| | + DIS | 47.06 | 77.43 | 85.22 | 2.0 | 10.34 |
| | + DualIS | 47.95 | **78.36** | 86.00 | 2.0 | 9.70 |
| | + DualDIS | 47.95 | 78.35 | 86.00 | 2.0 | 9.70 |

Table 3: Text-to-Video Retrieval performance on MSVD. Best in **Bold** and the second best is underlined.

on MSR-VTT, AcvitityNet, and MSVD are presented in Tables 1 to 3. Due to the limitation of

| Methods | Normalization | R@1↑ | R@5↑ | R@10↑ | MdR↓ | MnR↓ |
|---|---|---|---|---|---|---|
| MSCOCO (5k split) | | | | | | |
| ViLT | | 40.4 | 70.0 | 81.1 | - | - |
| ALIGN | | 45.6 | 69.8 | 78.6 | - | - |
| CODIS | | 53.9 | 79.5 | 87.1 | - | - |
| ALBEF | | 60.7 | 84.3 | 90.5 | - | - |
| CLIP | | 30.31 | 54.74 | 66.12 | 4.0 | 25.39 |
| | + IS | 35.15 | 60.71 | 71.20 | 3.0 | 21.69 |
| | + DIS | 35.16 | 60.70 | 71.19 | 3.0 | 21.69 |
| | + DualIS | **37.93** | **63.36** | **73.37** | 3.0 | 21.23 |
| | + DualDIS | 37.92 | 63.35 | **73.37** | 3.0 | **21.17** |
| Oscar | | 52.50 | 80.03 | 87.96 | 1.0 | **10.68** |
| | + IS | 52.77 | 80.03 | 87.99 | 1.0 | 10.94 |
| | + DIS | 53.47 | 80.02 | 87.69 | 1.0 | 12.34 |
| | + DualIS | 53.86 | 80.39 | 88.09 | 1.0 | 11.83 |
| | + DualDIS | **53.91** | **80.57** | **88.10** | 1.0 | 11.59 |
| Flickr30k | | | | | | |
| ViLT | | 55.0 | 82.5 | 89.8 | - | - |
| UNITER | | 68.7 | 89.2 | 93.9 | - | - |
| ALIGN | | 75.7 | 93.8 | 96.8 | - | - |
| CODIS | | 79.7 | 94.8 | 97.3 | - | - |
| ALBEF | | 85.6 | 97.5 | 98.9 | - | - |
| CLIP | | 58.98 | **83.48** | 90.14 | 1.0 | **6.04** |
| | + IS | 57.78 | 83.40 | 90.16 | 1.0 | 6.20 |
| | + DIS | 59.02 | 83.40 | 90.10 | 1.0 | 6.04 |
| | + DualIS | 58.02 | 83.40 | **90.22** | 1.0 | 6.20 |
| | + DualDIS | 59.02 | 83.42 | 90.10 | 1.0 | 6.04 |
| Oscar | | 71.60 | 91.50 | **94.96** | 1.0 | 4.24 |
| | + IS | 72.26 | 91.74 | 94.88 | 1.0 | 4.26 |
| | + DIS | 72.26 | 91.74 | 94.88 | 1.0 | 4.26 |
| | + DualIS | 73.00 | 91.74 | 94.80 | 1.0 | **4.19** |
| | + DualDIS | 73.00 | 91.70 | 94.78 | 1.0 | 4.21 |

Table 4: Text-to-Image Retrieval performance on MSCOCO (5k split) and Flickr30k. Best in **Bold** and the second best is underlined. Due to the limitation of the computational resources, we only use 20% of data from the training data to construct the dual banks.

| Methods | Normalization | R@1↑ | R@5↑ | R@10↑ | MdR↓ | MnR↓ |
|---|---|---|---|---|---|---|
| AudioCaps | | | | | | |
| MoEE | | 23.00 | 55.70 | 71.00 | 4.0 | 16.30 |
| MMT | | 36.10 | 72.00 | 84.50 | 2.3 | 7.50 |
| AR-CE | | 22.23 | 54.49 | 70.54 | 5.0 | 15.89 |
| | + IS | 23.19 | 55.96 | 70.05 | 4.0 | 20.44 |
| | + DIS | 23.19 | 55.93 | 70.05 | 4.0 | 20.45 |
| | + DualIS | 24.04 | **56.84** | 71.03 | 4.0 | 18.44 |
| | + DualDIS | 24.04 | 56.81 | 71.03 | 4.0 | 18.44 |
| CLOTHO | | | | | | |
| MoEE | | 6.00 | 20.80 | 32.30 | 23.0 | 60.20 |
| MMT | | 6.50 | 21.60 | 66.90 | 23.0 | 67.70 |
| AR-CE | | 6.27 | 22.32 | 33.30 | 23.0 | 58.95 |
| | + IS | 6.83 | 23.50 | 35.04 | 22.0 | 56.52 |
| | + DIS | 6.83 | 23.46 | 34.97 | 22.0 | 56.61 |
| | + DualIS | 7.06 | 24.42 | 36.29 | 21.0 | 54.12 |
| | + DualDIS | **7.12** | **24.50** | 36.17 | 21.0 | **54.85** |

Table 5: Text-to-Audio Retrieval performance on AudioCaps and CLOTHO. Best in **Bold** and the second best is underlined.

space, the results for DiDemo and video-to-text retrieval results on MSR-VTT, ActivityNet, and MSVD are deferred to the Appendix. Moreover, we compare our results with state-of-the-art meth-

ods in video retrieval, *i.e.*, FSE (Zhang et al., 2018), HiT (Liu et al., 2021b), RoME (Rony et al., 2022), Frozen (Bain et al., 2021), DiscreteCodebook (Liu et al., 2022a), VCM (Cao et al., 2022), Center-CLIP (Zhao et al., 2022), Align&Tell (Wang et al., 2022b), CLIP2TV (Gao et al., 2021), X-Pool (Gorti et al., 2022), TS2-Net (Liu et al., 2022b), and CAMoE (Cheng et al., 2021). Notably, our approach, DBNORM, outperforms previous normalization methods by a significant margin on all four benchmarks. Specifically, with DualIS and DualDIS, the retrieval performance of CLIP4Clip yields the best performance, achieving R@1 of 45.00 and R@5 of 72.50. Similar observations can be obtained on other benchmarks. However, it is important to note that using the activation sets (DualDIS) may have an adverse effect on retrieval performance, possibly due to the biased construction of these sets.

**Text-image retrieval**. Quantitative results are shown in Table 4 and Table 10 in the Appendix. Similar observations can be obtained on MSCOCO and Flickr30k. DBNorm (DualIS and DualDIS) effectively improves R@1 compared with IS and DIS. We also compare our results with ALBEF (Li et al., 2021), ViLT (Kim et al., 2021b), ALIGN (Jia et al., 2021), UNITER (Chen et al., 2020b), and CODIS (Duan et al., 2022).

**Text-audio retrieval**. Results are presented in Table 5 and Table 11 in the Appendix. We observed that employing DualIS and DualDIS leads to significant improvements in R@1 for text-to-audio retrieval on two benchmarks. Specifically, R@1 on AudioCaps is increased to 24.04 with DualIS and DualDIS. Besides, we compare with MoEE (Miech et al., 2020), MMT (Gabeur et al., 2020).

### 3.3 Qualitative Results

To qualitatively validate the effectiveness of our proposed methods, we present examples of video-to-text and text-to-video retrieval on MSR-VTT in Figure 2 and Figure 5 in the Appendix, respectively. The retrieval results demonstrate that our proposed DualIS and DualDIS, leveraging gallery and query banks, exhibit better performance and robustness compared to IS and DIS. Specifically, Figure 2 (a) highlights the capability of DualIS and DualDIS to distinguish three distinct characters, while Figure 2 (b) showcases a specific case where the vanilla CLIP4Clip and CLIP4Clip with IS/DIS fails to capture the presence of a man wearing a black shirt in

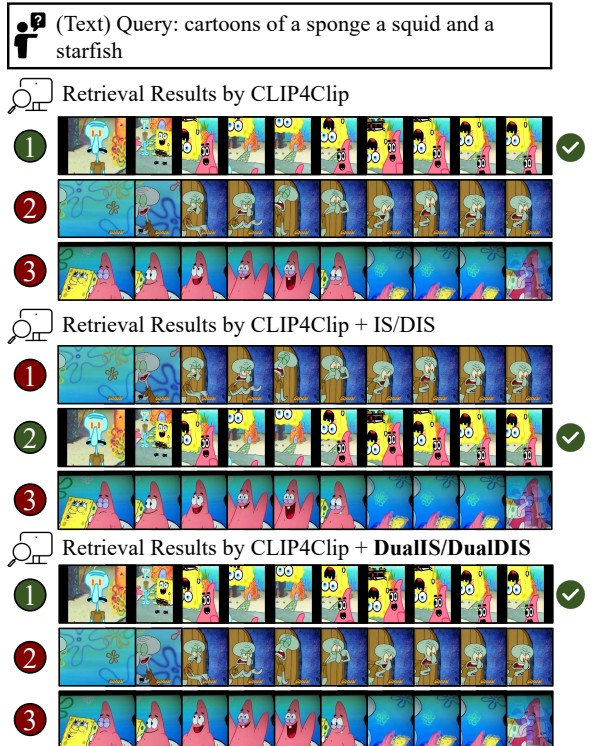

(a) Top-3 Text-to-Video retrieval examples.

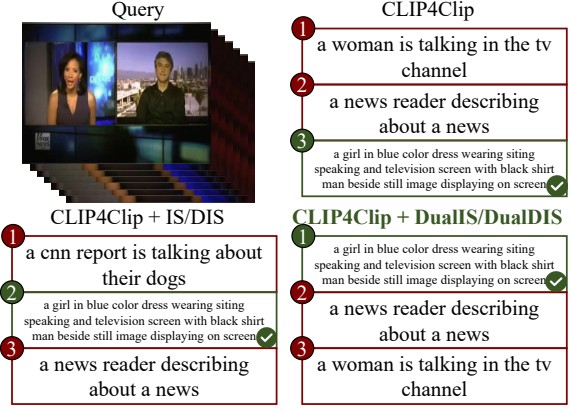

(b) Top-3 Video-to-Text retrieval examples.

Figure 2: Retrieval examples on MSR-VTT. More examples are presented in Figure 5 due to the limitation of space.

the background. In contrast, DualIS and DualDIS excel in capturing such intricate details, thereby yielding superior retrieval performance.

### 3.4 DBNORM

In this section, we answer several research questions of DBNORM on MSR-VTT and ActivityNet with CLIP4Clip. Due to the limitation of space, the questions on the sensitivity to $\beta_1$, $\beta_2$, and $k$ in DualDIS, the relationship between skewness and performance, aggregation methods, compar-

| Normalization | MSRVTT | | | | | ActivityNet | | | | MSCOCO | | Best |
|---|---|---|---|---|---|---|---|---|---|---|---|---|
| | CE+ | TT-CE+ | CLIP4Clip | CLIP2Video | X-CLIP | CE+ | TT-CE+ | CLIP4Clip | X-CLIP | CLIP | Oscar | |
| | 1.38 | 1.28 | 1.13 | 0.84 | 1.24 | 0.94 | 0.76 | 0.83 | 0.98 | 2.71 | 0.55 | 0 |
| + IS | 0.82 | 0.34 | **0.18** | 0.32 | 0.74 | 0.67 | 0.51 | 0.55 | 0.57 | 0.90 | 0.24 | 1 |
| + DIS | 0.83 | 0.34 | **0.18** | 0.33 | 0.74 | 0.68 | 0.52 | 0.42 | 0.44 | 0.90 | **0.22** | 2 |
| + DualIS | **0.37** | 0.33 | 0.57 | **0.26** | **0.70** | 0.54 | **0.37** | **0.36** | 0.42 | **0.42** | 0.31 | **7** |
| + DualDIS | **0.37** | **0.28** | 0.57 | **0.26** | **0.70** | 0.50 | 0.40 | 0.46 | 0.43 | 0.43 | 0.29 | 5 |

Table 6: The hubness (skewness score) on text-video/image retrieval with CE+, TT-CE+, CLIP4Clip, CLIP2Video, X-CLIP, CLIP, and Oscar is better reduced after applying our proposed DualIS and DualDIS than IS and DIS. The **Lower** is better. Best in **Bold** and the second best is underlined. The hubness scores on other benchmarks, methods, and image/video/audio-text retrieval are deferred to Table 13 in the Appendix due to the limitation of space.

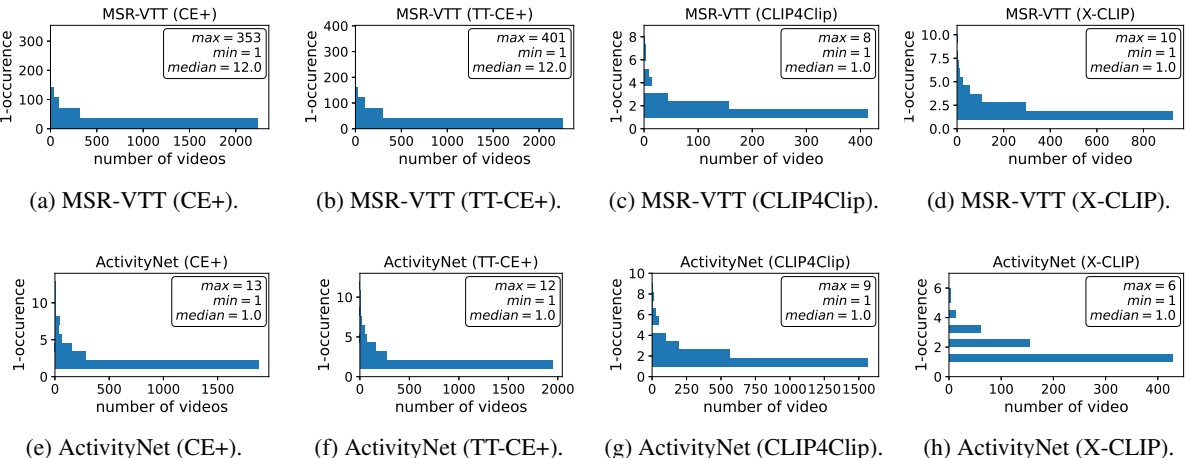

(a) MSR-VTT (CE+).    (b) MSR-VTT (TT-CE+).    (c) MSR-VTT (CLIP4Clip).    (d) MSR-VTT (X-CLIP).

(e) ActivityNet (CE+).    (f) ActivityNet (TT-CE+).    (g) ActivityNet (CLIP4Clip).    (h) ActivityNet (X-CLIP).

Figure 3: **Hubness is prevalent across different methods, datasets, and tasks**. These figures illustrate the distribution of the number of times each (test) gallery video was retrieved by (test) queries. Columns (different models): Retrieval distributions for CE, TT-CE+, CLIP4Clip, and X-CLIP. Rows (different datasets): Retrieval distributions for the same model on MSR-VTT and ActivityNet. The statistics (maximum, minimum, and median values) of 1-occurrence are shown in the figures. More illustrations are deferred to the Appendix (Figure 6).

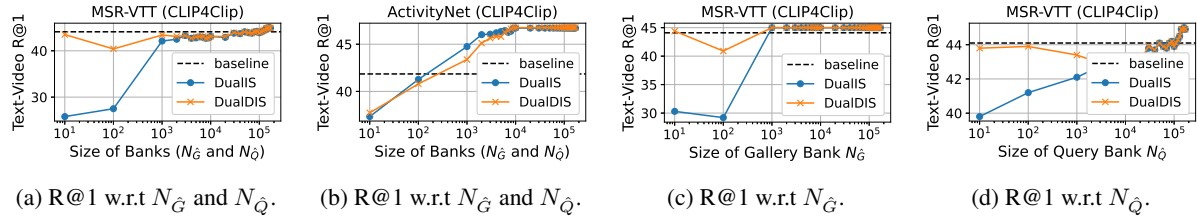

(a) R@1 w.r.t $N_{\hat{G}}$ and $N_{\hat{Q}}$.    (b) R@1 w.r.t $N_{\hat{G}}$ and $N_{\hat{Q}}$.    (c) R@1 w.r.t $N_{\hat{G}}$.    (d) R@1 w.r.t $N_{\hat{Q}}$.

Figure 4: Text-to-Video Retrieval R@1 w.r.t the size of gallery and query banks on MSR-VTT and ActivityNet using DualIs and DualDIS with CLIP4Clip.

ison with GC and CSLS, and the computational complexity are presented in the Appendix.

**RQ1: Can DBNORM alleviate hubness?** First, to illustrate the presence of the hubness problem, we present 1-occurrence in text-to-video/image/audio retrieval in Figure 3 and Figure 6 in the Appendix. Through the visualization across various benchmarks and methods, we consistently observe that a small subset of videos/images/audio is retrieved by multiple times,

resulting in a negative impact on performance, which empirically validates the prevalence of hubness. To quantitatively evaluate hubness, following Radovanovic et al. (2010), we report the skewness (details are presented in Appendix B.5) of text-to-video/image/audio and video/image/audio-to-text retrieval in Table 6 and Table 13 in the Appendix. The skewness is better reduced after employing DualIS and DualDIS than IS and DIS, proving the effectiveness of our methods in alleviating hubness.

**RQ2: How much data is desired in the banks?** To address this question, we conduct experiments on scaling the size of query and gallery banks by uniform sampling. The results of R@1 for text-video retrieval using DualIS and DualDIS are presented in Figures 4 and 7. Our observations indicate that as the size of the two banks increases, the performance improves. However, even with a relatively small number of samples in the query and gallery banks, we still observe satisfactory performance. Moreover, we examined the individual impact of the query and gallery bank sizes by independently sampling them at different scales. The results demonstrate that the size of the query bank has a greater influence on performance compared to the gallery bank, although a bigger gallery bank also leads to better performance.

## 4 Conclusion

In this work, we addressed the issue of hubness in cross-modal retrieval through a post-processing approach. First, we theoretically proved that hubs exhibit high similarity with data from both the query and gallery modalities. Then, motivated by our theoretical results, we proposed a novel post-processing method, Dual bank Normalization, along with two novel methods, *i.e.*, Dual Inverted Softmax and Dual Dynamic Inverted Softmax, which leveraged gallery and query banks to reduce similarities with hubs and improve similarities with non-hubs gallery data. Finally, with extensive experiments across a range of tasks, models, and benchmarks, we demonstrated the superiority of our proposed methods over previous methods in addressing hubness and improving performance.

## Limitations

First, our proposed method, DBNORM, is capable of tackling hubness in a range of cross-modal and single-modal retrieval tasks. However, in this study, we focused on evaluating DBNORM in the context of text-video, text-image, and text-audio retrieval, while excluding audio-image, audio-text, and audio-image retrieval tasks. It would be interesting to test the empirical efficiency of our methods on these tasks. Second, we observe that the performance improvement of DBNORM compared to single bank normalization is not significant on certain benchmarks, such as CLOTHO and MSVD. However, DBNORM has demonstrated satisfactory performance on MSR-VTT, ActivityNet,

MSCOCO, and AudioCaps datasets. Furthermore, we found that the hubness score (the skewness score) does not have an absolute correlation with the retrieval performance, as a low hubness score can still result in poor retrieval performance. In the future, it would be interesting to improve the robustness of bank-based normalization techniques and explore alternative effective metrics for modeling the relationship between hubness (skewness) and retrieval performance.

## Acknowledgments

We first sincerely thank all the reviewers and chairs for their efforts in helping improve our paper's presentation and organization. Next, we thank Tamer for the computational support and Charlie for reviewing the paper. Last but not least, we thank all the collaborators, friends, and computer science administrators for their support in Spring 2023.

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

# A    Related work

In this section, we present prior work from the literature that lies in cross-modal retrieval and the hubness problem, which are the two most related areas to our work.

**Cross-modal Retrieval.** Cross-modal retrieval methods aim to learn a common representation space, where the similarity between samples from different modalities can be directly measured. Early methods for cross-modal retrieval include Gaussian Mixture Models (Owens et al., 2016) modelling translation via EM (Croitoru et al., 2021), CCA  (Mithun et al., 2018), and KCCA (Patrick et al., 2021).

Inspired by the tremendous success of deep learning (Devlin et al., 2019; He et al., 2016), numerous methods have been proposed for image-text retrieval (Radford et al., 2021), video-text retrieval (Luo et al., 2022a), and audio-text retrieval (Oncescu et al., 2021). With the rise of self-supervised pretraining methods (Brown et al., 2020; Devlin et al., 2019; Radford et al., 2019), vision-language pretraining (Gan et al., 2020; Li et al., 2020b; Singh et al., 2022) on large-scale unlabeled cross-modal data has shown promising performance in various tasks, such as image retrieval (Liu et al., 2021a), image captioning (Jiang et al., 2019), and video retrieval (Park et al., 2022b). Inspired by this, recent works have attempted to pretrain or fine-tune cross-modal retrieval models, *e.g.*, image-text retrieval (Radford et al., 2021; Li et al., 2020b), video-text retrieval (Chen et al., 2020a; Cheng et al., 2021; Gao et al., 2021; Gorti et al., 2022; Lei et al., 2021; Ma et al., 2022; Park et al., 2022a; Wang et al., 2022a,b; Zhao et al., 2022), and audio-text retrieval (Koepke et al., 2022) in an end-to-end manner.

Another line of research has been focused on improving the effectiveness of retrieval, including k-d trees (Bellman, 2015), re-ranking (Zhong et al., 2017; Miech et al., 2021), query expansion (Chen et al., 2020a), vector compression schemes based on binary codes (Su et al., 2019; Liong et al., 2017) and quantization (Gong et al., 2013) that help address the curse of dimensionality (Keogh and Mueen, 2017). In addition, extracting additional information from noisy data (Hu et al., 2021), incorporating more datasets (Yu et al., 2023), and mitigating the domain adaptation problem (Kim et al., 2021a) have been other solutions for cross-modal retrieval.

Instead of improving cross-modal retrieval through better representation, retrieval techniques, or more data, we aim to advance cross-modal retrieval by addressing *the hubness problem* in a post-processing manner. This problem has been empirically demonstrated to be prevalent among various cross-modal retrieval tasks and datasets, as shown in Figure 3 and Figure 6 in the Appendix.

**The Hubness problem.** This problem was initially characterized by Radovanovic et al. (2010). They noticed that the distribution becomes considerably skewed as dimensionality increases under commonly used assumptions, leading to the emergence of hubs. Hubs are points with very high k-occurrences that effectively represent "popular" nearest neighbors. In other words, the distribution of "k-occurrences" (the number of times a point appears in the k nearest neighbors of other points) skews heavily to the right. As an inherent property of data distributions in high-dimensional vector space, it might cause the degradation of the retrieval (Bogolin et al., 2022) and bilingual word translation performance (Huang et al., 2020, 2019; Smith et al., 2017).

In the past few decades, algorithms for mitigating the hubness problem can be categorized into three classes. The first line of research focuses on rescaling the similarity space to symmetrize nearest neighbor relations (Schnitzer et al., 2012), including local (Jégou et al., 2007; Zelnik-Manor and Perona, 2004) and global (Schnitzer et al., 2012) scaling. Another line of research has focused on mitigating the hub tendency of centroids of the data through Laplacian-based kernels (Suzuki et al., 2012) and centering (Suzuki et al., 2013; Hara et al., 2015). Later, while CENT (Hara et al., 2015) has better complexity, Bogolin et al. (2022) show that CENT is not effective in terms of retrieval performance.

The last line, which is also closest to our paper, is the normalization of similarity by matching queries with a set of data points (Bogolin et al., 2022; Dinu et al., 2015; Lample et al., 2018; Smith et al., 2017). Dinu et al. (2015) first proposed a globally-corrected (GC) method to take into account the proximity distribution of potential neighbors across many mapped vectors to tackle the hubness problem. Following that, Lample et al. (2018) proposed cross-domain similarity local scaling (CSLS) using a bipartite neighborhood graph. Next, inverted softmax (IS) (Smith et al., 2017)

and QBNorm (DIS) (Bogolin et al., 2022) were proposed to normalize query similarities by considering the similarity between a query bank and the gallery points.

In contrast to IS and QBNorm, which are the two most related works to ours, our DBNORM is built upon a detailed *theoretical analysis* (Section 2.2) and normalizes similarity considering points from both modalities, as motivated by our theoretical analysis. By constructing banks from two modalities, DBNORM achieves better performance without access to any additional data on several different benchmarks across various cross-modal retrieval tasks.

**Hubness and similarity concentration.** It is known that the problem of hubness is related to concentration, the tendency of pairwise similarities between elements in a set to converge to a constant as the dimensionality of the space increases (Radovanovic et al., 2010). Later, Radovanović et al. (2010) show that it also holds for the widely employed cosine similarity as the expectation of pairwise similarities becomes constant and the standard deviation converges to 0.

## B  Experiments

To demonstrate the empirical efficiency of our DB-NORM, we test it alongside DualIS and DualDIS on five video-text retrieval methods: CE+ (Liu et al., 2019), TT-CE+ (Croitoru et al., 2021), CLIP4Clip (Luo et al., 2022a), X-CLIP (Ma et al., 2022), and CLIP2Video (Park et al., 2022a), two image-text retrieval methods: CLIP (Radford et al., 2021) and Oscar (Li et al., 2020b), and one audio-text retrieval method: AR-CE (Koepke et al., 2022).

### B.1  Dataset details

Experiments are conducted on eight cross-modal benchmarks, including four video-text retrieval benchmarks (MSR-VTT (Xu et al., 2016), MSVD (Chen and Dolan, 2011), ActivityNet (Fabian Caba Heilbron and Niebles, 2015), and DiDemo (Hendricks et al., 2017)), two image-text retrieval benchmarks (MSCOCO (Lin et al., 2014) and Flickr30k (Plummer et al., 2017)), and two audio-text retrieval benchmarks (AudioCaps (Kim et al., 2019) and CLOTHO (Drossos et al., 2020)). The details of the datasets are shown below,

- **MSR-VTT** (Xu et al., 2016) contains around 10k videos, each with 20 captions. For text-video retrieval, following prior works (Liu et al., 2019; Croitoru et al., 2021; Luo et al., 2022a; Ma et al., 2022; Park et al., 2022a), we use the official split (full) and the 1k-A split. The full split contains 2,990 videos for testing and 497 for validation while the 1k-A split contains 1,000 videos for testing and around 9,000 for training.

- **MSVD** (Chen and Dolan, 2011) has 1,970 videos and around 80k captions. The results are reported on the standard split used in prior works (Liu et al., 2019; Croitoru et al., 2021; Luo et al., 2022a; Park et al., 2022a) which consists of 1,200 videos for training, 100 for validation and 670 for testing.

- **ActivityNet** (Fabian Caba Heilbron and Niebles, 2015) contains 20k videos and has around 100K descriptive sentences. The videos are extracted from YouTube. We use a paragraph video retrieval as defined in prior works (Liu et al., 2019; Croitoru et al., 2021; Luo et al., 2022a; Park et al., 2022a). We report results on the val1 split while the training split consists of 10,009 videos, while there are 4,917 videos for testing.

- **DiDemo** (Hendricks et al., 2017) includes over 10,000 25-30 second long personal videos with over 40,000 localized text descriptions. Videos are split into training (8,395), validation (1,065), and testing (1,004) sets.

- **MSCOCO** (Lin et al., 2014) consists of 123k images with 5 captions for each sentence. We use the 5k split for evaluation.

- **Flickr30k** (Plummer et al., 2017) dataset contains 31,000 images collected from Flickr, together with 5 reference sentences provided by human annotators.

- **AudioCaps** (Kim et al., 2019) dataset which comprises sounds with event descriptions. We use the same setup as prior work (Koepke et al., 2022) where 49,291 samples are used for training, 428 for validation, and 816 for testing.

- **CLOTHO** (Drossos et al., 2020) consists of 4,981 audio samples of 15 to 30 seconds in duration and 24,905 captions of eight to 20 words in length (five captions for each).

| Methods | Normalization | Text-to-Video Retrieval | | | | | Video-to-Text Retrieval | | | | |
|---|---|---|---|---|---|---|---|---|---|---|---|
| | | R@1↑ | R@5↑ | R@10↑ | MdR↓ | MnR↓ | R@1↑ | R@5↑ | R@10↑ | MdR↓ | MnR↓ |
| MSR-VTT (full split) | | | | | | | | | | | |
| RoME | | 10.7 | 29.6 | 41.2 | 17.0 | - | - | - | - | - | - |
| Frozen | | 32.5 | 61.5 | 71.2 | - | - | - | - | - | - | - |
| CE+ | | 12.62 | 33.87 | 46.38 | 12.0 | 74.99 | 19.83 | 46.72 | 60.94 | 6.0 | 29.42 |
| | + IS | 13.31 | 35.00 | 47.46 | 12.0 | 74.16 | 23.68 | 52.25 | 65.59 | 5.0 | 24.75 |
| | + DIS | 13.31 | 35.00 | 47.46 | 12.0 | 74.16 | 23.68 | 52.25 | 65.59 | 5.0 | 24.75 |
| | + DualIS | **14.88** | **37.36** | **50.00** | **11.0** | **70.13** | **25.48** | **56.68** | **69.89** | **4.0** | **21.76** |
| | + DualDIS | **14.88** | **37.36** | **50.00** | **11.0** | **70.13** | **25.48** | **56.68** | **69.89** | **4.0** | **21.76** |
| TT-CE+ | | 14.61 | 37.81 | 50.78 | 10.0 | 63.31 | 24.48 | 54.11 | 67.59 | 5.0 | 20.22 |
| | + IS | 16.58 | 40.75 | 53.44 | 9.0 | 60.52 | 29.40 | 60.10 | 72.11 | 3.0 | 16.56 |
| | + DIS | 16.59 | 40.73 | 53.44 | 9.0 | 60.51 | 26.69 | 56.49 | 69.06 | 4.0 | 18.90 |
| | + DualIS | **17.06** | **41.63** | **54.25** | 8.0 | 60.10 | **29.83** | **62.04** | **73.68** | 3.0 | **15.70** |
| | + DualDIS | 17.02 | 41.60 | 54.23 | 8.0 | **60.01** | 27.53 | 57.39 | 69.77 | 4.0 | 18.63 |
| MSR-VTT (1k split) | | | | | | | | | | | |
| DiscreteCodebook | | 43.4 | 72.3 | 81.2 | - | 14.8 | 42.5 | 71.2 | 81.1 | - | 12.0 |
| VCM | | 43.8 | 71.0 | - | 2.0 | 14.3 | 45.1 | 72.3 | 82.3 | 2.0 | 10.7 |
| CenterCLIP | | 44.2 | 71.6 | 82.1 | 2.0 | 15.1 | 42.8 | 71.7 | 82.2 | 2.0 | 10.9 |
| Align&Tell | | 45.2 | 73.0 | 82.9 | 2.0 | - | 43.4 | 70.9 | 81.8 | 2.0 | - |
| CLIP2TV | | 45.6 | 71.1 | 80.8 | 2.0 | 15.0 | 43.9 | 70.9 | 82.2 | 2.0 | 12.0 |
| X-Pool | | 46.9 | 72.8 | 82.2 | 2.0 | 14.3 | - | - | - | - | - |
| TS2-Net | | 47.0 | 74.5 | 83.8 | 2.0 | 13.0 | 45.3 | 74.1 | 83.7 | 2.0 | 9.2 |
| CAMoE | | 47.3 | 74.2 | 84.5 | 2.0 | 11.9 | 49.1 | 74.3 | 84.3 | 2.0 | 9.9 |
| CLIP4Clip | | 44.10 | 71.70 | 81.40 | 2.0 | 15.51 | 42.09 | 71.24 | 81.23 | 2.0 | 12.01 |
| | + IS | 44.20 | 71.70 | 81.60 | 2.0 | 15.64 | 44.86 | 72.04 | 82.02 | 2.0 | **11.56** |
| | + DIS | 44.20 | 71.70 | 81.60 | 2.0 | 15.64 | 44.86 | 72.04 | 82.11 | 2.0 | 11.61 |
| | + DualIS | **45.00** | **72.50** | **82.10** | 2.0 | **15.32** | **45.45** | **73.02** | 81.42 | 2.0 | **11.56** |
| | + DualDIS | **45.00** | **72.50** | **82.10** | 2.0 | **15.32** | **45.45** | **73.02** | 81.42 | 2.0 | 11.59 |
| CLIP2Video | | 46.00 | 71.60 | 81.60 | 2.0 | 14.51 | 43.87 | 72.73 | 82.51 | 2.0 | 10.20 |
| | + IS | 47.00 | 72.80 | 82.10 | 2.0 | 13.91 | 46.15 | **72.63** | **81.92** | 2.0 | 11.15 |
| | + DIS | 47.00 | 72.80 | 82.10 | 2.0 | 13.91 | 46.15 | 72.53 | **81.92** | 2.0 | 11.14 |
| | + DualIS | **47.20** | **73.20** | **82.30** | 2.0 | **13.90** | 46.74 | 72.53 | 81.82 | 2.0 | 10.93 |
| | + DualDIS | **47.20** | 72.70 | **82.30** | 2.0 | **13.90** | 46.74 | 72.43 | 81.82 | 2.0 | **10.90** |
| X-CLIP | | 46.30 | 74.00 | 83.40 | 2.0 | **12.80** | 44.81 | 73.69 | 82.39 | 2.0 | 10.99 |
| | + IS | 48.60 | 74.10 | **84.10** | 2.0 | 13.35 | 46.69 | 73.99 | 83.28 | 2.0 | 10.52 |
| | + DIS | 48.60 | 74.10 | **84.10** | 2.0 | 13.35 | 46.69 | 73.89 | 83.28 | 2.0 | 10.52 |
| | + DualIS | **48.80** | **74.30** | **84.10** | 2.0 | 13.30 | 46.88 | 74.38 | 83.38 | 2.0 | **10.44** |
| | + DualDIS | 48.70 | **74.30** | **84.10** | 2.0 | 13.30 | 46.88 | 74.28 | 83.38 | 2.0 | **10.44** |

Table 7: Retrieval performance on MSR-VTT (full split and 1k split). Best in **Bold** and the second best is underlined.

## B.2 Experimental Details

The experiments are conducted using the PyTorch framework (Paszke et al., 2019). We utilize the following models and their respective weights: CE+ (Model Weights: MSR-VTT, MSVD, DiDeMo, and ActivityNet); TT-CE+ (Model Weights: MSR-VTT, MSVD, DiDeMo, and ActivityNet); CLIP2Video (Model Weights: MSR-VTT and MSVD); CLIP (Model Weights: MSCOCO and Flickr30k); Oscar (Model Weights: MSCOCO and Flickr30k); AR-CE (Model Weights: AudioCaps and CLOTHO) as they have released official code and weights. However, since CLIP4Clip and X-CLIP do not provide their trained models, we train

the models on a single A100 card using the hyper-parameters recommended in their papers. We set $k = 1$ in constructing activation sets.

## B.3 More Quantitative Results

**Text-video retrieval**. The video-to-text results on MSR-VTT, ActivityNet, and MSVD are presented in Tables 7 to 9, the results on DiDeMo are in Table 12. Similar results can be observed. Our proposed DualIS and DualDIS outperform the previous methods, *i.e.*, IS and DIS, by a large margin across different benchmarks.

**Text-image retrieval**. The image-to-text results on MSCOCO and Flickr30k are shown in Table 10.

**Text-audio retrieval**. The audio-to-text results

| Methods | Normalization | Video-to-Text Retrieval | | | | |
|---|---|---|---|---|---|---|
| | | R@1↑ | R@5↑ | R@10↑ | MdR↓ | MnR↓ |
| FSE | | 12.6 | 33/2 | 77.6 | 12.0 | - |
| VCM | | 42.6 | 74.9 | 86.2 | 2.0 | 6.4 |
| CenterCLIP | | 44.5 | 75.7 | 86.2 | 2.0 | 6.5 |
| Align&Tell | | 43.5 | 73.6 | 98.3 | 2.0 | - |
| CAMoE | | 49.9 | 77.4 | - | - | - |
| CE+ | | 18.51 | 47.85 | 63.94 | 6.0 | 23.06 |
| | + IS | 19.36 | 50.13 | 65.87 | 5.0 | 20.81 |
| | + DIS | 19.52 | 49.87 | 65.81 | 5.0 | 21.01 |
| | + DualIS | **21.92** | **53.49** | **68.25** | 5.0 | **17.57** |
| | + DualDIS | 21.72 | 52.69 | 67.62 | 5.0 | 18.16 |
| TT-CE+ | | 22.49 | 56.38 | 72.67 | 4.0 | 13.90 |
| | + IS | 23.18 | 57.01 | 73.60 | 4.0 | 13.30 |
| | + DIS | 23.43 | 57.05 | 73.48 | 4.0 | 13.48 |
| | + DualIS | **27.46** | **61.13** | **76.71** | 4.0 | **11.00** |
| | + DualDIS | 27.11 | 60.79 | 75.86 | 4.0 | 11.42 |
| CLIP4Clip | | 41.62 | 74.11 | 86.12 | 2.0 | 6.81 |
| | + IS | 46.23 | 76.72 | 87.26 | 2.0 | 6.46 |
| | + DIS | 46.26 | 76.48 | 87.16 | 2.0 | 6.48 |
| | + DualIS | 46.59 | **78.04** | **88.15** | 2.0 | **6.05** |
| | + DualDIS | **46.73** | 77.90 | 88.06 | 2.0 | 6.05 |
| X-CLIP | | 45.20 | 76.07 | 86.57 | 2.0 | 6.40 |
| | + IS | **51.13** | 78.95 | 88.14 | 1.0 | 5.62 |
| | + DIS | 50.52 | 78.60 | 87.84 | 1.0 | 5.71 |
| | + DualIS | 50.92 | **79.42** | **89.36** | 1.0 | **5.32** |
| | + DualDIS | 50.22 | 79.12 | 88.62 | 1.0 | 5.44 |

Table 8: Retrieval performance on ActivityNet. Best in **Bold** and the second best is underlined.

| Methods | Normalization | Video-to-Text Retrieval | | | | |
|---|---|---|---|---|---|---|
| | | R@1↑ | R@5↑ | R@10↑ | MdR↓ | MnR↓ |
| FSE | | 16.7 | 43.1 | 88.4 | 7.0 | - |
| CenterCLIP | | 63.5 | 86.4 | 92.6 | 1.0 | 3.8 |
| Align&Tell | | 61.8 | 87.5 | 92.7 | 1.0 | - |
| CE+ | | 22.84 | 49.85 | 61.49 | 6.0 | 33.96 |
| | + IS | 26.27 | 54.48 | 65.97 | 4.0 | 27.69 |
| | + DIS | 24.63 | 52.99 | 64.63 | 5.0 | 32.57 |
| | + DualIS | **29.85** | **59.40** | **67.46** | 4.0 | **25.41** |
| | + DualDIS | 25.67 | 53.58 | 65.07 | 5.0 | 32.26 |
| TT-CE+ | | 25.22 | 55.07 | 64.63 | 4.0 | 29.94 |
| | + IS | 24.63 | 52.39 | 65.07 | 4.0 | 27.31 |
| | + DIS | 23.43 | 55.22 | 65.37 | 4.0 | 28.25 |
| | + DualIS | **28.06** | **57.46** | **67.61** | 4.0 | **25.64** |
| | + DualDIS | 24.18 | 55.97 | 65.82 | 4.0 | 28.16 |
| CLIP4Clip | | 63.13 | 79.40 | 85.37 | 1.0 | 11.02 |
| | + IS | 67.61 | **84.48** | **89.70** | 1.0 | **7.61** |
| | + DIS | 64.48 | 81.49 | 87.16 | 1.0 | 11.10 |
| | + DualIS | **67.76** | 84.18 | 89.25 | 1.0 | 8.07 |
| | + DualDIS | 64.78 | 81.64 | 87.16 | 1.0 | 11.12 |
| CLIP2Video | | 62.09 | 83.13 | 89.40 | 1.0 | 7.73 |
| | + IS | 66.57 | 82.84 | 88.36 | 1.0 | 7.94 |
| | + DIS | 62.39 | 82.84 | 88.51 | 1.0 | 8.73 |
| | + DualIS | **69.70** | **86.27** | **89.70** | 1.0 | **6.35** |
| | + DualDIS | 63.13 | 83.58 | 89.40 | 1.0 | 7.47 |
| X-CLIP | | 65.67 | 83.73 | 89.85 | 1.0 | **8.15** |
| | + IS | 64.63 | 81.64 | 87.16 | 1.0 | 9.84 |
| | + DIS | 64.78 | 82.84 | 88.51 | 1.0 | 8.46 |
| | + DualIS | **67.91** | 83.58 | **88.96** | 1.0 | 8.33 |
| | + DualDIS | 66.27 | **83.73** | 88.81 | 1.0 | 8.27 |

Table 9: Retrieval performance on MSVD. Best in **Bold** and the second best is underlined.

| Methods | Normalization | Image-to-Text Retrieval | | | | |
|---|---|---|---|---|---|---|
| | | R@1↑ | R@5↑ | R@10↑ | MdR↓ | MnR↓ |
| MSCOCO | | | | | | |
| ViLT | | 56.5 | 82.6 | 89.6 | - | - |
| ALIGN | | 58.6 | 83.0 | 89.7 | - | - |
| CODIS | | 71.5 | 91.1 | 95.5 | - | - |
| ALBEF | | 77.6 | 94.3 | 97.2 | - | - |
| CLIP | | 50.02 | 74.82 | 83.34 | 1.0 | 9.23 |
| | + IS | 50.68 | 74.80 | 83.10 | 1.0 | 9.39 |
| | + DIS | 50.68 | 74.80 | 83.10 | 1.0 | 9.39 |
| | + DualIS | 53.26 | **77.00** | 85.26 | 1.0 | 8.44 |
| | + DualDIS | **53.32** | 76.96 | **85.30** | 1.0 | **8.32** |
| Oscar | | 66.74 | 89.98 | 94.98 | 1.0 | 2.98 |
| | + IS | 68.36 | 90.16 | 95.32 | 1.0 | 2.91 |
| | + DIS | 69.48 | 90.54 | 95.04 | 1.0 | 2.99 |
| | + DualIS | 70.72 | 91.06 | 95.66 | 1.0 | 2.81 |
| | + DualDIS | **70.93** | **91.44** | **95.84** | 1.0 | **2.76** |
| Flickr30k | | | | | | |
| ViLT | | 73.2 | 93.6 | 96.5 | - | - |
| UNITER | | 83.6 | 95.7 | 97.7 | - | - |
| ALIGN | | 88.6 | 98.7 | 99.7 | - | - |
| CODIS | | 91.7 | 99.3 | 99.8 | - | - |
| ALBEF | | 95.9 | 99.8 | 100.0 | - | - |
| CLIP | | 78.10 | 94.90 | **98.10** | 1.0 | 1.98 |
| | + IS | 77.80 | 92.90 | 97.40 | 1.0 | 2.15 |
| | + DIS | 77.90 | 92.90 | 97.40 | 1.0 | 2.14 |
| | + DualIS | **81.60** | 95.40 | 97.90 | 1.0 | **1.92** |
| | + DualDIS | 81.50 | 95.40 | 97.90 | 1.0 | **1.92** |
| Oscar | | 86.30 | 96.80 | 98.60 | 1.0 | 1.58 |
| | + IS | 87.10 | 97.70 | 99.10 | 1.0 | 1.49 |
| | + DIS | 87.10 | 97.70 | 99.10 | 1.0 | 1.49 |
| | + DualIS | 87.10 | 97.60 | 99.30 | 1.0 | 1.49 |
| | + DualDIS | 87.10 | 97.60 | 99.30 | 1.0 | 1.49 |

Table 10: Image-to-Text Retrieval performance on MSCOCO (5k split) and Flickr30k. Best in **Bold** and the second best is underlined. Due to the limitation of the computational resources, we only use 20% of data from the training data to construct the dual banks.

| Methods | Normalization | Audio-to-Text Retrieval | | | | |
|---|---|---|---|---|---|---|
| | | R@1↑ | R@5↑ | R@10↑ | MdR↓ | MnR↓ |
| AudioCaps | | | | | | |
| MoEE* | | 26.60 | 59.30 | 73.50 | 4.0 | 15.60 |
| MMT* | | 39.60 | 76.80 | 86.70 | 2.0 | 6.50 |
| AR-CE | | 24.02 | 56.00 | 71.81 | 4.0 | 16.91 |
| | + IS | 29.90 | 61.64 | 74.75 | 4.0 | **13.95** |
| | + DIS | 29.90 | 61.27 | 74.75 | 4.0 | 15.01 |
| | + DualIS | 30.02 | **62.13** | **75.12** | 4.0 | 14.80 |
| | + DualDIS | **30.15** | 61.76 | **75.12** | 4.0 | 14.85 |
| CLOTHO | | | | | | |
| MoEE* | | 7.20 | 22.10 | 33.20 | 22.7 | 71.80 |
| MMT* | | 6.30 | 22.80 | 33.30 | 22.3 | 67.30 |
| AR-CE | | 7.27 | 23.35 | 35.12 | 21.0 | 74.68 |
| | + IS | 8.52 | 23.64 | 37.70 | 20.0 | 72.99 |
| | + DIS | 8.33 | 23.35 | 36.84 | 20.0 | 73.35 |
| | + DualIS | **9.09** | **25.55** | **38.28** | 19.0 | **71.33** |
| | + DualDIS | 8.13 | 23.64 | 37.04 | 20.0 | 72.71 |

Table 11: Audio-to-Text Retrieval performance on AudioCaps and CLOTHO. Best in **Bold** and the second best is underlined.

on AudioCaps and CLOTHO are presented in Table 11.

## B.4 More Qualitative Results

This section presents retrieval results for text-to-video and video-to-text retrieval tasks, as illustrated

| Methods | Normalization | Text-to-Video Retrieval | | | | | Video-to-Text Retrieval | | | | |
|---|---|---|---|---|---|---|---|---|---|---|---|
| | | R@1 ↑ | R@5 ↑ | R@10 ↑ | MdR ↓ | MnR ↓ | R@1 ↑ | R@5 ↑ | R@10 ↑ | MdR ↓ | MnR ↓ |
| S2VT | | 11.9 | 33.6 | 76.5 | 13.0 | - | 13.2 | 33.6 | 76.5 | 15.0 | - |
| FSE | | 13.9 | 36.0 | 78.9 | 11.0 | - | 13.1 | 33.9 | 78.0 | 12.0 | - |
| CE+ | | 18.22 | 42.63 | 56.08 | 8.0 | 42.05 | 18.82 | 42.73 | 55,78 | 8.0 | 37.27 |
| | + IS | 21.22 | **46.31** | 59.56 | **7.0** | **37.06** | 20.72 | 45.72 | **59.86** | 7.0 | **33.84** |
| | + DIS | 21.02 | **46.31** | 59.56 | 7.0 | 37.37 | 20.72 | 45.72 | 59.76 | 7.0 | 33.87 |
| | + DualIS | **21.81** | 46.12 | **59.66** | 7.0 | 37.07 | **20.92** | 45.62 | 59.66 | 7.0 | **33.84** |
| | + DualDIS | 21.71 | **46.31** | 59.56 | 7.0 | 37.37 | 20.72 | 45.82 | 59.56 | 7.0 | 33.90 |
| TT-CE+ | | 22.31 | 49.20 | 62.15 | 6.0 | 31.18 | 21.41 | 47.11 | 60.66 | 6.0 | 29.67 |
| | + IS | 23.80 | **51.29** | **65.04** | 5.0 | **28.21** | 24.20 | 51.59 | 65.54 | 5.0 | 26.09 |
| | + DIS | 23.71 | **51.29** | 64.84 | 5.0 | 28.40 | 24.30 | 51.49 | 65.54 | 5.0 | 26.23 |
| | + DualIS | **25.60** | 50.30 | 64.44 | 5.0 | 28.49 | 24.40 | **52.59** | **66.33** | 5.0 | **25.01** |
| | + DualDIS | **25.60** | 50.40 | 64.44 | 5.0 | 28.88 | **24.60** | 52.49 | 66.24 | 5.0 | 25.08 |

Table 12: Retrieval performance on DiDeMo. Best in **Bold** and the second best is underlined. S2VT (Venugopalan et al., 2015) and FSE (Zhang et al., 2018) are two representative methods on DiDeMo.

in Figures 2 and 5, respectively. Upon observing both text-to-video and video-to-text retrieval scenarios, a notable observation emerges regarding the susceptibility of IS and DIS to be misled by the query bank as shown in Figure 5. In contrast, our proposed methods, DualIS and DualDIS, capitalize on the utilization of both query and gallery banks, leading to an improved retrieval performance achieved by effectively reducing the similarity of hubs and preserving the similarity of non-hubs.

### B.5 DBNORM

**Continued RQ1: Can DBNORM alleviate hubness? (Empirical Observation of Hubness in Cross-Modal Retrieval)** We first illustrate the hubness phenomenon across several benchmarks and methods of cross-modal retrieval in Figure 3 and Figure 6. We notice that, hubness is prevalent across different methods, datasets, and tasks as the queries frequently retrieve a small number of gallery data.

Moreover, to demonstrate how severe the hubness problem is in a specific benchmark and method, following Radovanovic et al. (2010), we employ skewness of the k-occurrences distribution, with $k = 10$. Specifically, the skewness of the k-occurrences distribution is defined as

$$S_{N_k} = \frac{\mathbb{E}_{\mathbf{x}}[N_k(\mathbf{x}) - \mu_{N_k}]^3}{\sigma_{N_k}^3}, \quad (3)$$

where $\mu_{N_k}$ and $\sigma_{N_k}$ are the mean and standard deviation of $N_k$. $N_k(\mathbf{x})$ refers to the $k$-occurrence distribution, given by

$\sum_{i \in [N_g]} p_{i,k}(\mathbf{x}) = \sum_{i \in [N_g]} \mathbf{I}(\mathbf{x}$ is in the $k$ nearest neighbours of the $i$-th gallery data) and $\mathbf{I}(cond)$ is the indicator function. Positive skewness indicates that the distribution is right-tailed and higher skewness scores mean a severer hubness problem occurs. As shown in Tables 6 and 13, the hubness score consistently exhibits a relatively high value across various methods and benchmarks. However, upon employing our DualIS and DualDIS, we observed a notable reduction in hubness scores across different benchmarks. This reduction serves as empirical evidence supporting the effectiveness and efficiency of our proposed methods in addressing the hubness issue.

**Continued RQ2: How much data is desired in the banks?** The results of R@1 for text-video retrieval using DualIS and DualDIS are presented in Figures 4 and 7. Our observations indicate that as the size of the two banks increases, the performance improves. owever, even with a relatively small number of samples in the query and gallery banks, we still achieve satisfactory performance. Moreover, we examined the individual impact of the query and gallery bank sizes by independently sampling them at different scales. The results demonstrate that the size of the query bank has a greater influence on performance compared to the gallery bank, although a bigger gallery bank also leads to better performance.

**RQ3: Is DBNORM sensitive to $\beta_1$ and $\beta_2$?** In order to investigate the sensitivity of DBNORM to $\beta_1$ and $\beta_2$, we conducted an evaluation as shown in Figure 8. Our findings indicate that the performance of DualDIS demonstrates limited sensitivity

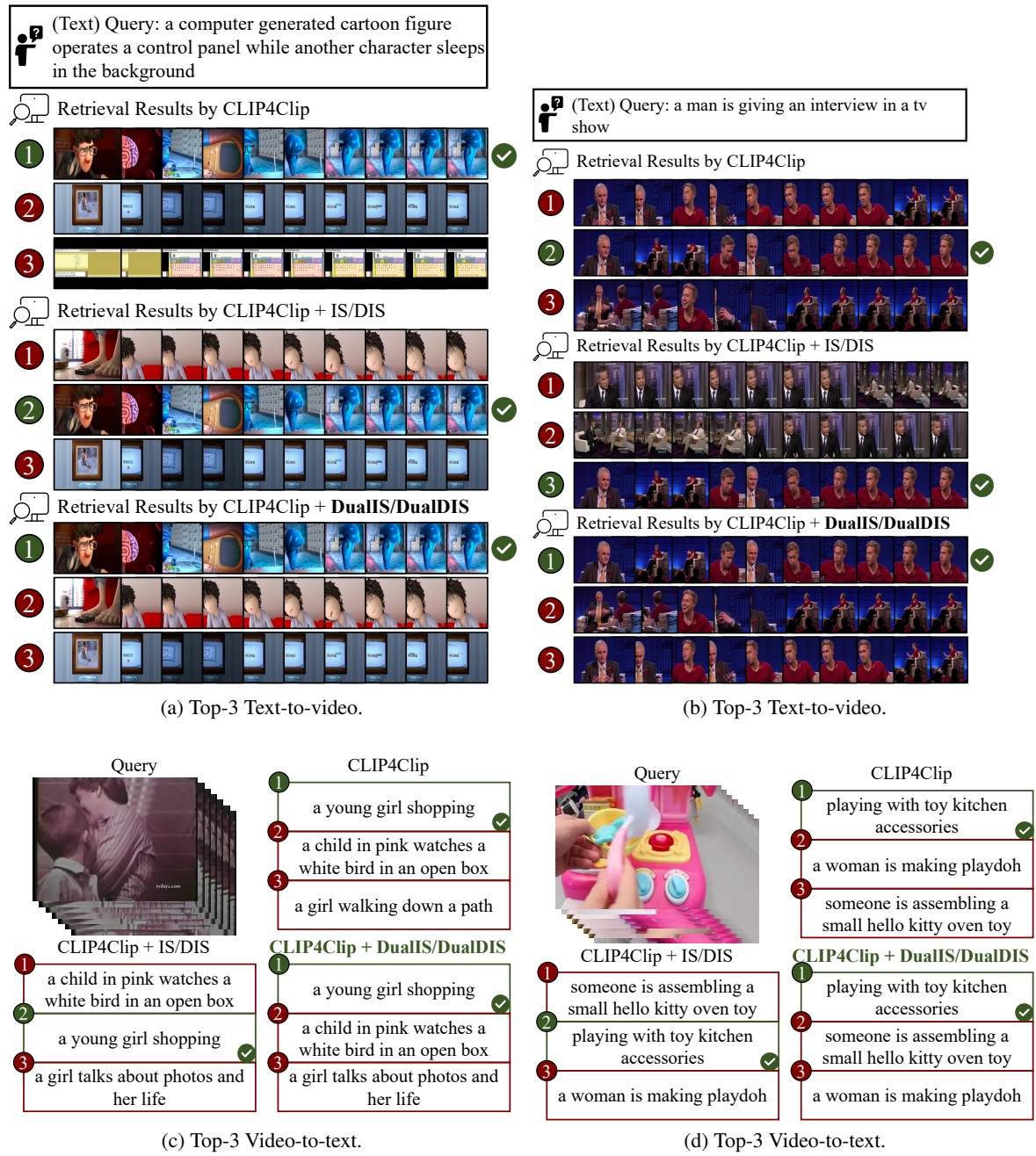

(a) Top-3 Text-to-video.

(b) Top-3 Text-to-video.

(c) Top-3 Video-to-text.

(d) Top-3 Video-to-text.

Figure 5: Top-3 Text-to-video and Video-to-text retrieval results on MSR-VTT.

to changes in both $\beta_1$ and $\beta_2$. Adjusting these hyperparameters within a reasonable range does not result in significant variations in the overall performance.

**RQ4: Is DualDIS sensitive to the Top-k hyperparameter?** We conducted experiments to observe the influence of k in the Top-k selection used to construct the gallery activation sets. The results are shown in Figure 9. We observed that choosing $k = 1$ offers a good trade-off between performance and computational efficiency. Therefore, we used $k = 1$ for all reported experiments.

**RQ5: Is hubness score related to performance?** Our current findings reveal an inverse relationship between the hubness score and performance. Specifically, we observed that as the hubness score decreases, the performance tends to improve. This aligns with the expectation that lower hubness scores indicate a reduced concentration of nearest neighbors, leading to improved retrieval performance. However, it is important to note that in certain cases (X-CLIP on MSVD), the relationship between the hubness score and retrieval performance may differ from this trend.

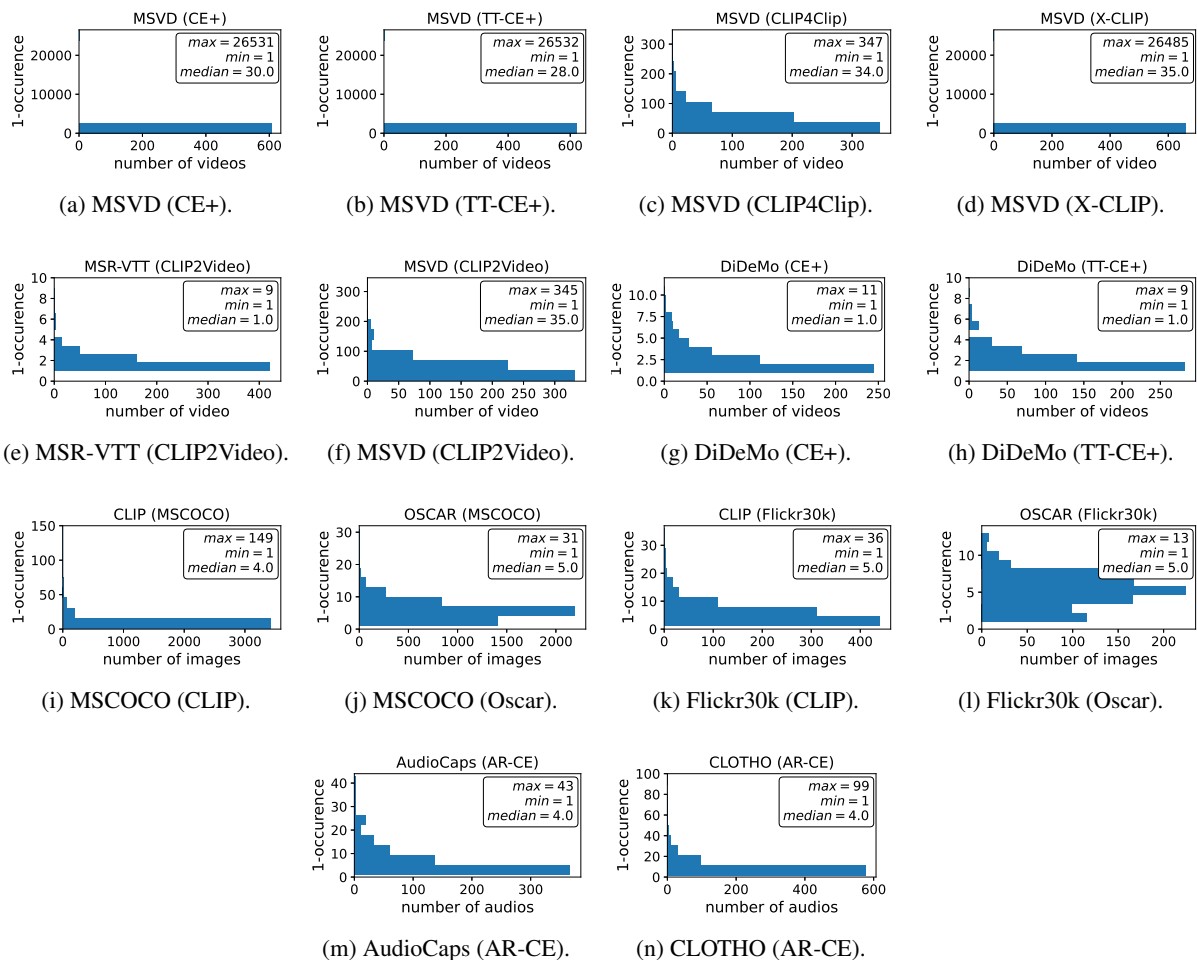

Figure 6: **Hubness is prevalent across different methods, datasets, and tasks**. These figures illustrate the distribution of the number of times each (test) gallery video/image/audio was retrieved by (test) set queries. In all visualization, severe hubness can be observed, as a small number of galleries are retrieved disproportionately often, which might reduce the retrieval performance.

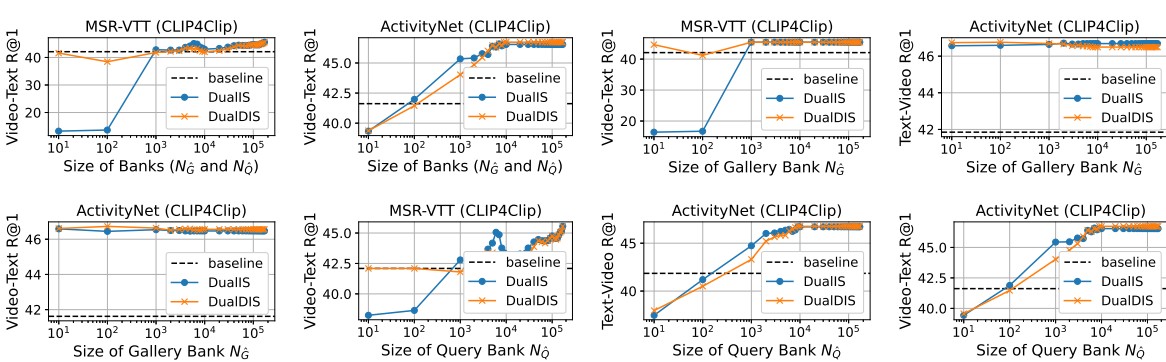

Figure 7: Text-Video Retrieval R@1 w.r.t the size of gallery and query banks on MSR-VTT and ActivityNet using DualIs and DualDIS with CLIP4Clip.

This can be attributed to potential biases present in the dataset and the models, which can influence the observed patterns. Therefore, we should carefully interpret the hubness score as the sole indicator

of retrieval performance, particularly when dataset biases are significant. Further investigations are needed to gain a comprehensive understanding of the interplay between the hubness score and re-

| Normalization | MSRVTT | | | | | ActivityNet | | | | MSCOCO | | Best |
|---|---|---|---|---|---|---|---|---|---|---|---|---|
| | CE+ | TT-CE+ | CLIP4Clip | CLIP2Video | X-CLIP | CE+ | TT-CE+ | CLIP4Clip | X-CLIP | CLIP | Oscar | |
| *Text-to-Video/Image/Audio* | | | | | | | | | | | | |
| | 1.38 | 1.28 | 1.13 | 0.84 | 1.24 | 0.94 | 0.76 | 0.83 | 0.98 | 2.71 | 0.55 | 0 |
| + IS | 0.82 | 0.34 | **0.18** | 0.32 | 0.74 | 0.67 | 0.51 | 0.55 | 0.57 | 0.90 | 0.24 | 1 |
| + DIS | 0.83 | 0.34 | **0.18** | 0.33 | 0.74 | 0.68 | 0.52 | 0.42 | 0.44 | 0.90 | **0.22** | 1 |
| + DualIS | **0.37** | 0.33 | 0.57 | **0.26** | **0.70** | 0.54 | **0.37** | **0.36** | **0.42** | **0.42** | 0.31 | 7 |
| + DualDIS | **0.37** | **0.28** | 0.57 | **0.26** | **0.70** | **0.50** | 0.40 | 0.46 | 0.43 | 0.43 | 0.29 | 5 |
| *Video/Image/Audio-to-Text* | | | | | | | | | | | | |
| | 3.65 | 4.02 | 2.13 | 2.13 | 2.66 | 1.06 | 0.70 | 1.11 | 1.58 | 3.78 | 1.45 | 0 |
| + IS | **2.29** | **2.34** | **0.43** | **0.28** | 1.58 | 0.63 | 0.56 | 0.77 | 0.44 | 2.21 | 1.08 | 4 |
| + DIS | 2.54 | 2.50 | 0.47 | 0.31 | 1.61 | 0.65 | 0.57 | 0.58 | 0.56 | 2.21 | 1.17 | 0 |
| + DualIS | 2.48 | 2.38 | 1.12 | **0.28** | **1.46** | **0.30** | **0.27** | **0.32** | **0.37** | 1.53 | **0.98** | 7 |
| + DualDIS | 2.51 | 2.47 | 1.16 | 0.31 | 1.48 | 0.34 | 0.29 | 0.40 | **0.37** | **1.50** | 1.06 | 2 |

| Normalization | MSVD | | | | | DiDeMo | | Flickr30K | | AudioCaps | CLOTHO | Best |
|---|---|---|---|---|---|---|---|---|---|---|---|---|
| | CE+ | TT-CE+ | CLIP4Clip | CLIP2Video | X-CLIP | CE+ | TT-CE+ | CLIP | Oscar | AR-CE | AR-CE | |
| *Text-to-Video/Image/Audio* | | | | | | | | | | | | |
| | 7.95 | 7.95 | 0.83 | 0.84 | **0.65** | 1.23 | 0.86 | 2.78 | 0.32 | 0.22 | 1.09 | 1 |
| + IS | **7.88** | 7.92 | 0.42 | 0.80 | 1.87 | **0.44** | 0.39 | 4.06 | **0.01** | 0.02 | 0.68 | 4 |
| + DIS | 7.92 | 7.92 | 0.42 | 0.80 | 1.87 | 0.46 | 0.40 | **2.81** | **0.01** | **0.01** | 0.68 | 4 |
| + DualIS | 7.93 | 7.92 | **0.37** | **0.20** | 0.68 | 0.53 | 0.33 | 4.14 | **0.01** | 0.15 | **0.46** | 5 |
| + DualDIS | 7.93 | 7.92 | **0.37** | 0.21 | 0.68 | 1.13 | **0.32** | **2.81** | **0.01** | 0.15 | 0.55 | 5 |
| *Video/Image/Audio-to-Text* | | | | | | | | | | | | |
| | 3.18 | 4.18 | 4.54 | 6.02 | 4.47 | 1.01 | 0.98 | 2.23 | 1.16 | 1.65 | 2.43 | 0 |
| + IS | 3.45 | **3.06** | **2.97** | **2.78** | **3.84** | 0.53 | 0.72 | 1.62 | 1.02 | 1.00 | 2.03 | **4** |
| + DIS | 3.42 | 4.05 | 3.47 | 4.39 | 4.64 | **0.52** | 0.72 | 1.62 | 1.03 | **0.99** | 2.02 | 2 |
| + DualIS | **2.60** | 3.42 | 3.01 | 3.00 | 3.92 | 0.54 | **0.59** | 1.08 | **0.99** | 1.06 | **1.53** | 4 |
| + DualDIS | 3.43 | 4.04 | 3.60 | 4.23 | 4.57 | 0.56 | **0.59** | **1.06** | **0.99** | 1.04 | 1.60 | 3 |

Table 13: The hubness (skewness score) is significantly reduced after applying our proposed DualIS and DualDIS than IS and DIS Best in **Bold** and the second best is underlined.

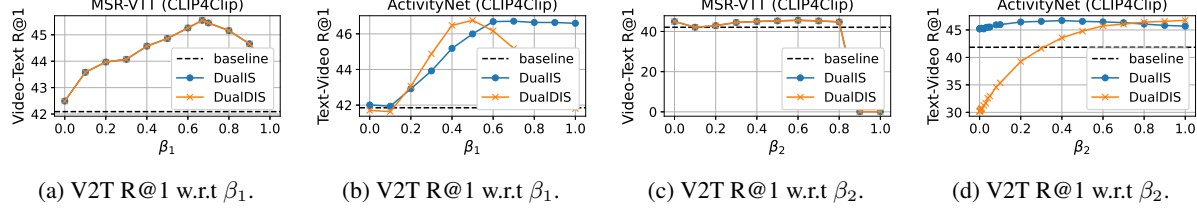

(a) V2T R@1 w.r.t $\beta_1$.  (b) V2T R@1 w.r.t $\beta_1$.  (c) V2T R@1 w.r.t $\beta_2$.  (d) V2T R@1 w.r.t $\beta_2$.

Figure 8: Video-to-Text Retrieval recall at 1 w.r.t $\beta_1$ and $\beta_2$ on CLIP4Clip and ActivityNet using DualIS and DualDIS.

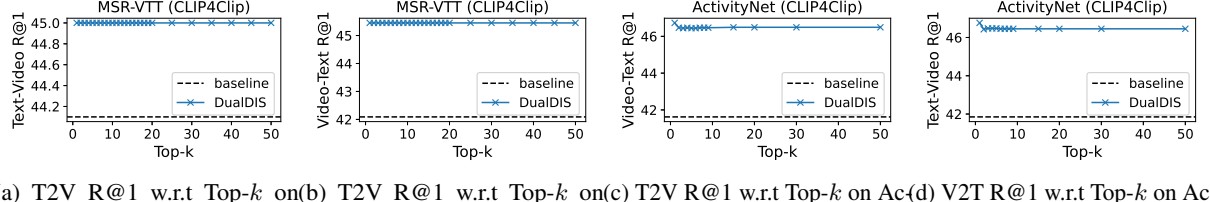

(a) T2V R@1 w.r.t Top-$k$ on MSR-VTT.  (b) T2V R@1 w.r.t Top-$k$ on MSR-VTT.  (c) T2V R@1 w.r.t Top-$k$ on ActivityNet.  (d) V2T R@1 w.r.t Top-$k$ on ActivityNet.

Figure 9: Video-Text Retrieval performance with respect to top-k hyperparameter in DualDIS.

trieval performance under different conditions and datasets.

**RQ6: How about aggreagating normalization results by adding instead of multiplying?** In Du-

alIS (Equation (1)) and DualDIS (Equation (2)), multiplication is employed to aggregate normalization results from the query and gallery banks. To explore alternative aggregation methods, we

| | Agg | Text-to-Video Retrieval | | | | | Video-to-Text Retrieval | | | | |
|---|---|---|---|---|---|---|---|---|---|---|---|
| | | R@1↑ | R@5↑ | R@10↑ | MdR↓ | MnR↓ | R@1↑ | R@5↑ | R@10↑ | MdR↓ | MnR↓ |
| MSR-VTT (1k) | | | | | | | | | | | |
| Clip4Clip | | 44.10 | 71.70 | 81.40 | 2.0 | 15.51 | 42.09 | 71.24 | 81.23 | 2.0 | 12.01 |
| + DualIS | + | 44.20 | 71.70 | 81.60 | 2.0 | 15.64 | 44.86 | 72.04 | 82.02 | 2.0 | 11.56 |
| + DualDIS | + | **45.00** | **72.50** | **82.10** | 2.0 | **15.32** | **45.45** | **73.02** | **81.42** | 2.0 | 11.59 |
| + DualIS | × | **45.00** | **72.50** | **82.10** | 2.0 | **15.32** | **45.45** | **73.02** | **81.42** | 2.0 | **11.56** |
| + DualDIS | × | **45.00** | **72.50** | **82.10** | 2.0 | **15.32** | **45.45** | **73.02** | **81.42** | 2.0 | 11.59 |
| ActivityNet | | | | | | | | | | | |
| CLIP4Clip | | 41.85 | 74.44 | 84.84 | 2.0 | 6.84 | 41.62 | 74.11 | 86.12 | 2.0 | 6.81 |
| + DualIS | + | 46.40 | **77.85** | 87.21 | 2.0 | 6.14 | 47.02 | 77.59 | 87.49 | 2.0 | 6.16 |
| + DualDIS | + | 46.26 | 77.78 | 86.90 | 2.0 | 6.11 | **47.09** | 77.21 | 87.38 | 2.0 | 6.31 |
| + DualIS | × | 46.71 | 77.47 | **87.35** | 2.0 | **6.01** | 46.59 | **78.04** | **88.15** | 2.0 | **6.05** |
| + DualDIS | × | **46.76** | 77.48 | 87.28 | 2.0 | **6.01** | 46.73 | 77.90 | 88.06 | 2.0 | **6.05** |

Table 14: Retrieval results on MSR-VTT and ActivityNet with DBNORM(DualIS) and DBNORM(DualDIS). "Agg" refers to the aggregation methods. "×" and "+" represent multiplying and addition.

| | QB | GB | w/o test | Text-to-Video Retrieval | | | | | | Video-to-Text Retrieval | | | | | |
|---|---|---|---|---|---|---|---|---|---|---|---|---|---|---|---|
| | | | | R@1↑ | R@5↑ | R@10↑ | MdR↓ | MnR↓ | Skewness↓ | R@1↑ | R@5↑ | R@10↑ | MdR↓ | MnR↓ | Skewness↓ |
| MSR-VTT (1k) | | | | | | | | | | | | | | | |
| Clip4Clip | | | | 44.10 | 71.70 | 81.40 | 2.0 | 15.51 | 1.38 | 42.09 | 71.24 | 81.23 | 2.0 | 12.01 | 3.65 |
| + GC (upper bound) | | ✓ | | 45.60 | 71.30 | 81.10 | 2.0 | 16.50 | 0.45 | 46.84 | 70.85 | 80.34 | 2.0 | 18.71 | 0.37 |
| + GC | | ✓ | ✓ | 40.20 | 65.10 | 75.60 | 2.0 | 24.52 | 3.33 | 40.14 | 67.33 | 76.59 | 2.0 | 15.58 | 4.71 |
| + CSLS (upper bound, $k = 1000$)) | ✓ | ✓ | | 42.70 | 70.70 | 80.70 | 2.0 | 15.93 | 0.81 | 42.39 | 70.85 | 80.43 | 2.0 | 12.84 | 0.88 |
| + CSLS with $k = 10$ | ✓ | ✓ | ✓ | 41.70 | 69.20 | 79.20 | 2.0 | 16.55 | 1.33 | 5.57 | 63.04 | 73.42 | 3.0 | 19.11 | 2.71 |
| + CSLS with $k = 100$ | ✓ | ✓ | ✓ | 43.10 | 70.60 | 80.40 | 2.0 | 16.02 | 0.92 | 41.11 | 69.86 | 79.25 | 2.0 | 13.93 | 1.43 |
| + CSLS with $k = 1000$ | ✓ | ✓ | ✓ | 42.70 | 71.00 | 80.70 | 2.0 | 16.01 | 0.90 | 42.19 | 70.36 | 80.43 | 2.0 | 13.11 | **0.92** |
| + IS | ✓ | | ✓ | 44.20 | 71.70 | 81.60 | 2.0 | 15.64 | 0.82 | 44.86 | 72.04 | 82.02 | 2.0 | **11.56** | 2.29 |
| + DIS | ✓ | | ✓ | 44.20 | 71.70 | 81.60 | 2.0 | 15.64 | 0.83 | 44.86 | 72.04 | 82.11 | 2.0 | 11.61 | 2.54 |
| + DualIS | ✓ | ✓ | ✓ | **45.00** | **72.50** | **82.10** | 2.0 | **15.32** | **0.37** | **45.45** | **73.02** | 81.42 | 2.0 | **11.56** | 2.48 |
| + DualDIS | ✓ | ✓ | ✓ | **45.00** | **72.50** | **82.10** | 2.0 | **15.32** | **0.37** | **45.45** | **73.02** | 81.42 | 2.0 | 11.59 | 2.51 |
| ActivityNet | | | | | | | | | | | | | | | |
| CLIP4Clip | | | | 41.85 | 74.44 | 84.84 | 2.0 | 6.84 | 0.94 | 41.62 | 74.11 | 86.12 | 2.0 | 6.81 | 1.06 |
| + GC (upper bound) | | ✓ | | 48.03 | 77.57 | 87.42 | 2.0 | 6.59 | 0.13 | 47.09 | 77.48 | 86.71 | 2.0 | 6.52 | 0.20 |
| + GC | | ✓ | ✓ | 33.89 | 63.22 | 76.74 | 3.0 | 9.91 | 2.19 | 32.28 | 61.18 | 75.39 | 3.0 | 10.11 | 56.88 |
| + CSLS (upper bound, $k = 4222$) | ✓ | ✓ | | 40.29 | 73.00 | 84.46 | 2.0 | 7.32 | 0.79 | 40.90 | 73.38 | 85.53 | 2.0 | 7.04 | 0.92 |
| + CSLS $k = 10$ | ✓ | ✓ | ✓ | 37.54 | 70.46 | 82.35 | 2.0 | 7.89 | 1.18 | 38.42 | 71.06 | 83.37 | 2.0 | 7.70 | 1.30 |
| + CSLS $k = 100$ | ✓ | ✓ | ✓ | 40.34 | 72.50 | 84.23 | 2.0 | 7.36 | 0.90 | 40.31 | 73.05 | 84.98 | 2.0 | 7.11 | 0.99 |
| + CSLS $k = 1000$ | ✓ | ✓ | ✓ | 40.15 | 72.93 | 84.41 | 2.0 | 7.34 | 0.81 | 40.74 | 73.38 | 85.53 | 2.0 | 7.04 | 0.92 |
| + CSLS $k = 4222$ | ✓ | ✓ | ✓ | 40.29 | 72.93 | 84.49 | 2.0 | 7.32 | 0.79 | 40.83 | 73.33 | 85.41 | 2.0 | 7.05 | 0.94 |
| IS | ✓ | | ✓ | 45.93 | 77.52 | 87.07 | 2.0 | 6.39 | 0.67 | 46.23 | 76.72 | 87.26 | 2.0 | 6.46 | 0.63 |
| + DIS | ✓ | | ✓ | 46.02 | 77.29 | 86.83 | 2.0 | 6.29 | 0.68 | 46.26 | 76.48 | 87.16 | 2.0 | 6.48 | 0.65 |
| + DualIS | ✓ | ✓ | ✓ | 46.71 | 77.47 | **87.35** | 2.0 | **6.01** | 0.54 | 46.59 | **78.04** | **88.15** | 2.0 | **6.05** | **0.30** |
| + DualIS | ✓ | ✓ | ✓ | **46.76** | **77.48** | 87.28 | 2.0 | **6.01** | 0.50 | **46.73** | 77.90 | 88.06 | 2.0 | **6.05** | 0.34 |

Table 15: Retrieval results on MSR-VTT and ActivityNet with GC, CSLS, IS, DIS, DBNORM(DualIS), and DBNORM(DualDIS). "QB", "GB", and "w/o test" refers to the query bank, gallery bank, and without the access to all test gallery data (only observing one gallery data at a time).

conducted experiments by replacing multiplication with addition. The retrieval results are presented in Table 14. It is observed that, for MSR-VTT, both multiplication and addition yield comparable retrieval performance, whereas for ActivityNet, multiplication outperforms addition, providing empirical evidence for the superiority of multiplication.

## B.6 Comparison with GC and CSLS

Globally-corrected (**GC**) (Dinu et al., 2015) and Cross-Domain Similarity Local Scaling (**CSLS**) (Lample et al., 2018) are two representative post-processing methods specifically designed

to address hubness in Natural Language Processing. Note that, GC and CSLS requires the access to all queries.

However, in real-world settings, queries do not always occur simultaneously, and it is common for methods not to have access to all queries at once. This aligns with the practical conditions where queries are generated in a sequential or asynchronous manner. To test the compatibility of GC and CSLS with such scenarios, we adapt GC and CSLS accordingly and conduct experiments on the MSR-VTT and ActivityNet datasets.

GC returns the gallery element **g** that has the highest rank for query **q**. The similarity is normal-

ized as follows,

$$\hat{s}_{\mathbf{q},\mathbf{g}_i} = s_{\mathbf{q},\mathbf{g}_i} - Rank(\mathbf{g}_i, Q, \mathbf{q}) , \qquad (4)$$

where $Rank(\mathbf{g}_i, Q, \mathbf{q})$ returns the rank of $\mathbf{q}$ in $Q$ when the query is $\mathbf{g}_i$ and $Q$ contains all the queries. To investigate the applicability of GC without accessing other test queries, we modify GC as follows,

$$\hat{s}_{\mathbf{q},\mathbf{g}_i} = s_{\mathbf{q},\mathbf{g}_i} - Rank(\mathbf{g}_i, \hat{Q} \cup \{\mathbf{q}\}, \mathbf{q}) , \qquad (5)$$

where $Rank(\mathbf{g}_i, \hat{Q} \cup \mathbf{q}, \mathbf{q})$ returns the rank of $\mathbf{q}$ in $\hat{Q} \cup \mathbf{q}$ when the query is $\mathbf{g}_i$.

CSLS utilizes a query bank that consists of all the test queries and a gallery bank. To adapt CSLS for use in our method, we modify it to incorporate (train or validation) queries and normalize the similarity $s_{\mathbf{q},\mathbf{g}_i}$ between a query $\mathbf{q}$ and a gallery point $\mathbf{g}_i$ as follows,

$$\hat{s}_{\mathbf{q},\mathbf{g}_i} = 2s_{\mathbf{q},\mathbf{g}_i} - \frac{1}{K}\sum_{i \in [K]} s_{\mathbf{q},\bar{\mathbf{g}}_i} - \frac{1}{K}\sum_{i \in [K]} s_{\mathbf{q},\bar{\mathbf{q}}_i} , \qquad (6)$$

where $\{\bar{\mathbf{g}}_i\}_{i \in [k]}$ and $\{\bar{\mathbf{q}}_i\}_{i \in [k]}$ are the $k$ gallery and query samples in banks that are most similar to query $\mathbf{q}$.

The quantitative results of GC and CSLS on MSR-VTT and ActivityNet are shown in Table 15. We denote GC and CSLS with access to test queries as GC (upperbound) and CSLS (upperbound) respectively. It is noteworthy that the retrieval performance significantly drops without access to all test queries, and the performance of CSLS is proportional to the hyperparamter $k$.

Considering the unsatisfactory retrieval performance of GC and CSLS when lacking access to all test queries, we exclude these two methods from our approach.

## C  Proofs

### C.1  Proof of Theorem 1

Consider two data points $\mathbf{x}_1$ and $\mathbf{x}_2$ sampled from $\mathcal{N}(\boldsymbol{\mu})$, which satisfy

$$\|\mathbf{x}_2 - \boldsymbol{\mu}\| - \|\mathbf{x}_1 - \boldsymbol{\mu}\| > 0 . \qquad (7)$$

The expected difference between the squared Euclidean distances from $\mathbf{x}_1$ and $\mathbf{x}_2$ to a point $\mathbf{x}$ sampled from the same distribution, is defined as

$$\Delta = \mathbb{E}_{\mathbf{x}}\left[\|\mathbf{x}_2 - \mathbf{x}\|^2 - \|\mathbf{x}_1 - \mathbf{x}\|^2\right] \\ = \mathbb{E}_{\mathbf{x}}\left[\|\mathbf{x}_2 - \mathbf{x}\|^2\right] - \mathbb{E}_{\mathbf{x}}\left[\|\mathbf{x}_1 - \mathbf{x}\|^2\right] . \qquad (8)$$

We notice that,

$$\mathbb{E}_{\mathbf{x}}\left[\|\mathbf{x}_2 - \mathbf{x}\|^2\right] \\ = \mathbb{E}_{\mathbf{x}}\left[\|\mathbf{x}_2 - \boldsymbol{\mu} - (\mathbf{x} - \boldsymbol{\mu})\|^2\right] \\ = \mathbb{E}_{\mathbf{x}}\left[\|\mathbf{x}_2 - \boldsymbol{\mu}\|^2\right] + \mathbb{E}\left[\|\mathbf{x} - \boldsymbol{\mu}\|^2\right] \\ \quad - 2\mathbb{E}\left[(\mathbf{x}_2 - \boldsymbol{\mu})^\top(\mathbf{x} - \boldsymbol{\mu})\right] \\ = \mathbb{E}\left[\|\mathbf{x}_2 - \boldsymbol{\mu}\|^2\right] + \mathbb{E}\left[\|\mathbf{x} - \boldsymbol{\mu}\|^2\right] .$$

The last equality is because the mean of $\mathbf{x}_2$ and $\mathbf{x}$ is $\boldsymbol{\mu}$.

Similarly, we have

$$\mathbb{E}\left[\|\mathbf{x}_1 - \mathbf{x}\|^2\right] \\ = \mathbb{E}\left[\|\mathbf{x}_1 - \boldsymbol{\mu}\|^2\right] + \mathbb{E}\left[\|\boldsymbol{\mu} - \mathbf{x}\|^2\right] . \qquad (9)$$

Next, we have,

$$\Delta = (\mathbb{E}\left[\|\mathbf{x}_2 - \boldsymbol{\mu}\|^2\right] + \mathbb{E}\left[\|\boldsymbol{\mu} - \mathbf{x}\|^2\right]) \\ \quad - (\mathbb{E}\left[\|\mathbf{x}_1 - \boldsymbol{\mu}\|^2\right] + \mathbb{E}\left[\|\boldsymbol{\mu} - \mathbf{x}\|^2\right]) \qquad (10) \\ = \|\mathbf{x}_2 - \boldsymbol{\mu}\|^2 - \|\mathbf{x}_1 - \boldsymbol{\mu}\|^2 > 0 .$$

Now, we have completed the proof.

### C.2  Proof of Theorem 2

The difference $\Delta$ is defined as,

$$\Delta = \mathbb{E}_{\mathbf{y},\mathbf{x}_1 \sim \mathcal{X},\mathbf{x}_2 \sim \mathcal{X}, \|\mathbf{x}_2 - \boldsymbol{\mu}_x\|^2 - \|\mathbf{x}_1 - \boldsymbol{\mu}_x\|^2 > 0}\left[\|\mathbf{x}_2 - \mathbf{y}\|^2 - \|\mathbf{x}_1 - \mathbf{y}\|^2\right] . \qquad (11)$$

We have,

$$\mathbb{E}\left[\|\mathbf{x}_2 - \mathbf{y}\|^2\right] \\ = \mathbb{E}\left[\|\mathbf{x}_2 - \boldsymbol{\mu}_x - (\mathbf{y} - \boldsymbol{\mu}_x)\|^2\right] \\ = \mathbb{E}\|\mathbf{x}_2 - \boldsymbol{\mu}_x\|^2 + \mathbb{E}\|\mathbf{y} - \boldsymbol{\mu}_x\|^2 \\ \quad - 2\mathbb{E}\left[(\mathbf{x}_2 - \boldsymbol{\mu}_x)^\top(\mathbf{y} - \boldsymbol{\mu}_x)\right] \\ = \mathbb{E}\|\mathbf{x}_2 - \boldsymbol{\mu}_x\|^2 + \mathbb{E}\|\mathbf{y} - \boldsymbol{\mu}_x\|^2$$

Similarly, we have,

$$\mathbb{E}\left[\|\mathbf{x}_1 - \mathbf{y}\|^2\right] = \mathbb{E}\|\mathbf{x}_1 - \boldsymbol{\mu}_x\|^2 + \mathbb{E}\|\mathbf{y} - \boldsymbol{\mu}_x\|^2 .$$

Inserting the above two equality into Eq. (11), we have

$$\Delta = \mathbb{E}\left[\|\mathbf{x}_2 - \mathbf{y}\|^2 - \|\mathbf{x}_1 - \mathbf{y}\|^2\right] \\ = \|\mathbf{x}_2 - \boldsymbol{\mu}_x\|^2 - \|\mathbf{x}_1 - \boldsymbol{\mu}_x\|^2 > 0 .$$

## C.3 Proof of Corollary 3

We only need to prove that if $\mathbb{E}\left[\|\mathbf{x}_2 - \mathbf{y}\|^2 - \|\mathbf{x}_1 - \mathbf{y}\|^2\right] > 0$, we have $\mathbb{E}\left[\|\mathbf{x}_2 - \mathbf{x}\|^2 - \|\mathbf{x}_1 - \mathbf{x}\|^2\right] > 0$, where $\mathbf{y} \sim \mathcal{Y}, \mathbf{x} \sim \mathcal{X}$.

As $\mathbb{E}\left[\|\mathbf{x}_2 - \mathbf{y}\|^2 - \|\mathbf{x}_1 - \mathbf{y}\|^2\right] > 0$, with the proof of Theorem 2, we have

$$\|\mathbf{x}_2 - \boldsymbol{\mu}_x\|^2 - \|\mathbf{x}_1 - \boldsymbol{\mu}_x\|^2 > 0.$$

In other words, as $\|\mathbf{x}_2 - \boldsymbol{\mu}_x\| + \|\mathbf{x}_1 - \boldsymbol{\mu}_x\| > 0$, we have

$$\|\mathbf{x}_2 - \boldsymbol{\mu}_x\| - \|\mathbf{x}_1 - \boldsymbol{\mu}_x\| > 0.$$

With Theorem 1, we have $\mathbb{E}\left[\|\mathbf{x}_2 - \mathbf{x}\|^2 - \|\mathbf{x}_1 - \mathbf{x}\|^2\right] > 0$.

Now, we complete the proof.

## C.4 Proof of Theorem 4

Similarly, we first present the following theorems.

**Theorem 5.** *Assuming that $\mathbf{x}_1$ and $\mathbf{x}_2$ are sampled from a symmetric distribution $\mathcal{X}$ on a hypersphere with mean $\boldsymbol{\mu}$, if $\mathbf{x}_1$ is closer to $\boldsymbol{\mu}$ than $\mathbf{x}_2$ (i.e., $\cos(\mathbf{x}_1, \boldsymbol{\mu}) - \cos(\mathbf{x}_2, \boldsymbol{\mu})$), then $\mathbf{x}_1$ is more likely to be a hub than $\mathbf{x}_2$ on the space $\mathcal{X}$.*

*Proof.* Consider two data points $\mathbf{x}_1$ and $\mathbf{x}_2$ sampled from a symmetric distribution on the surface of a hypersphere satisfying $\|\mathbf{x}_1\|_2 = \|\mathbf{x}_2\|_2 = r$, and the following holds true

$$\cos(\mathbf{x}_1, \boldsymbol{\mu}) - \cos(\mathbf{x}_2, \boldsymbol{\mu}) > 0. \tag{12}$$

where $\boldsymbol{\mu}$ is the center of the distribution. That means under the cosine metric, $\mathbf{x}_1$ is closer to the mean point than $\mathbf{x}_2$.

The expected difference between the cosine distances from $\mathbf{x}_1$ and $\mathbf{x}_2$ to a random point $\mathbf{x}$ sampled from the same distribution, is defined as

$$
\begin{aligned}
\Delta =& \mathbb{E}\left[\cos(\mathbf{x}_1, \mathbf{x}) - \cos(\mathbf{x}_2 - \mathbf{x})\right] \\
=& \frac{1}{r^2} \mathbb{E}\left[\mathbf{x}_1\mathbf{x}^\top - \mathbf{x}_2\mathbf{x}^\top\right] \\
=& \frac{1}{r^2} \mathbb{E}[\mathbf{x}_1 - \mathbf{x}_2]\boldsymbol{\mu}^\top \\
=& \cos(\mathbf{x}_1, \boldsymbol{\mu}) - \cos(\mathbf{x}_2, \boldsymbol{\mu}) > 0.
\end{aligned}
\tag{13}
$$

Now, we have completed the proof. $\square$

**Theorem 6.** *Assuming that $\mathbf{x}_1$ and $\mathbf{x}_2$ are sampled from a symmetric distribution with mean $\boldsymbol{\mu}_x$, if $\mathbf{x}_1$ is closer to $\boldsymbol{\mu}$ than $\mathbf{x}_2$ (i.e., $\cos(\mathbf{x}_1, \boldsymbol{\mu}) - \cos(\mathbf{x}_2, \boldsymbol{\mu})$), then $\mathbf{x}_1$ is more likely to be a hub than $\mathbf{x}_2$ on another space $\mathcal{Y}$.*

*Proof.* Consider two data points $\mathbf{x}_1$ and $\mathbf{x}_2$ sampled from a symmetric distribution $\mathcal{X}$ and a random point $\mathbf{y}$ sampled from a symmetric distribution $\mathcal{Y}$ on the surface of a hypersphere satisfying $\|\mathbf{x}_1\|_2 = \|\mathbf{x}_2\|_2 = \|\mathbf{y}\|_2 = r$, and the following holds true

$$\cos(\mathbf{x}_1, \boldsymbol{\mu}_x) - \cos(\mathbf{x}_2, \boldsymbol{\mu}_x) > 0. \tag{14}$$

where $\boldsymbol{\mu}_x$ is the center of the distribution $\mathcal{X}$. That means under the cosine metric, $\mathbf{x}_1$ is closer to the mean point than $\mathbf{x}_2$.

The expected difference between the cosine distances from $\mathbf{x}_1$ and $\mathbf{x}_2$ to the random point $\mathbf{y}$, is defined as

$$
\begin{aligned}
\Delta =& \mathbb{E}\left[\cos(\mathbf{x}_1, \mathbf{y}) - \cos(\mathbf{x}_2 - \mathbf{y})\right] \\
=& \frac{1}{r^2} \mathbb{E}\left[\mathbf{x}_1\mathbf{y}^\top - \mathbf{x}_2\mathbf{y}^\top\right] \\
=& \frac{1}{r^2} \mathbb{E}[\mathbf{x}_1 - \mathbf{x}_2](\boldsymbol{\mu}_y - \boldsymbol{\mu}_x)^\top \\
& + \frac{1}{r^2} \mathbb{E}[\mathbf{x}_1 - \mathbf{x}_2](\boldsymbol{\mu}_x)^\top \\
=& \cos(\mathbf{x}_1, \boldsymbol{\mu}_x) - \cos(\mathbf{x}_2, \boldsymbol{\mu}_x) > 0.
\end{aligned}
\tag{15}
$$

Now, we have completed the proof. $\square$

Combining the above two theorems and with the proof of Corollary 3, we have completed the proof.