# OpenReview forum: "Balance Act: Mitigating Hubness in Cross-Modal Retrieval with Query and Gallery Banks"
_EMNLP/2023/Conference — EMNLP 2023 Main_

### Official Review · Reviewer_GfvP · 2023-08-03

**Soundness:** 5

**Excitement:**

4: Strong: This paper deepens the understanding of some phenomenon or lowers the barriers to an existing research direction.

**Paper Topic And Main Contributions:**

This paper proposes Dual Bank Normalization (DBNorm), a general post-processing solution that addresses the hubness problem in cross-modal retrieval, which is a phenomenon where a small number of gallery data points are frequently retrieved, resulting in a decrease in retrieval performance. The main contributions of the paper are:
1. introducing DBNorm, a theory-grounded novel framework, as a general solution to mitigate the hubness problem in the context of cross-modal retrieval (CMR), along with two post-processing methods: Dual Inverted Softmax (DualIS) and Dual Dynamic Inverted Softmax (DualDIS),
2. demonstrate through experimental results on a diverse set of benchmarks that DBNorm outperforms previous methods in addressing hubness and improving retrieval performance.

**Reasons To Accept:**

1. The proposed DBNorm approach addresses a fundamental problem common to most CMR models and has the potential to improve the performance of cross-modal retrieval systems in general.
2. The paper provides a rigorous theoretical demonstration of the context of the problem and the motivation for the key idea of the solution (namely, normalizing on both query and gallery data). Similarly, the proposal of DualIS and DualDIS is well-motivated by prior work.
3. The paper provides quantitative experimental results of the method on 8 language-based benchmarks of different cross-modal modality settings (4 text-video retrieval benchmarks, 2 text-image retrieval benchmarks, and 2 audio-text retrieval benchmarks), and convincingly demonstrates that the proposed DBNorm approach outperforms previous normalization methods based on SOTA models in a variety of settings.

**Reasons To Reject:**

The designs of the main contributions appear to be a small increment from the previous research (Bogolin et al., 2022), which identified the hubness problem in CMR and proposed the single-bank normalization method DIS (on which DualDIS is based). This is partially mitigated by the theoretical foundation of DBNorm and the extensive experimental studies demonstrating superior performance, which improves the soundness of this work by providing a clear motivation.

**Reproducibility:**

5: Could easily reproduce the results.

**Reviewer Confidence:**

3: Pretty sure, but there's a chance I missed something. Although I have a good feel for this area in general, I did not carefully check the paper's details, e.g., the math, experimental design, or novelty.

---

> ### Author Rebuttal · Authors · 2023-08-24
>
> We sincerely thank you for your time, efforts, and your detailed and positive comments. We have carefully considered your comments and have provided responses to specific comments.
>
> **Reasons To Reject:**
>
> >We deeply appreciate your thorough feedback and would like to begin by highlighting our approach to mitigating the hubness problem [1,2]. Our initial step involves employing larger banks to normalize similarities, a strategy demonstrated to effectively reduce the hubness issue [1,2].
>
> >Building on this foundation, our contributions stem from two key directions:
> >
> >1. [**Theoretical foundation**] Similar to the query bank, the gallery bank holds the potential for identifying hubs that can contribute to hubness alleviation. This observation is underscored by Theorems 1 and 2, providing a theoretical basis for our approach. Moreover, our theoretical findings also lay the groundwork for QBNorm [2].
> >2. [**Novel framework**] In line with the theoretical insights, we propose DBNorm, supplemented by the DualIS and DualDIS. These techniques harness the power of dual banks to address the hubness problem with enhanced efficacy.
>
> > The fusion of these theoretical foundations and novel methodologies has enabled us to present a holistic solution to the hubness problem in CMR, thereby enhancing the robustness and applicability of our work.
>
> We sincerely thank you for your thoughtful comments and are open to further discussion on these matters.
>
> [1] Milos Radovanovic, Alexandros Nanopoulos, and Mirjana Ivanovi. 2010. Hubs in space: Popular nearest neighbors in high-dimensional data. Journal of Machine Learning Research, 11(86):2487–2531.
>
> [2] Simion-Vlad Bogolin, Ioana Croitoru, Hailin Jin, Yang Liu, and Samuel Albanie. 2022. Cross modal retrieval with querybank normalisation. In IEEE/CVF Conference on Computer Vision and Pattern Recognition, CVPR 2022, New Orleans, LA, USA, June 18-24, 2022, pages 5184–5195. IEEE.

---

### Official Review · Reviewer_htxD · 2023-08-04

**Soundness:** 4

**Excitement:**

4: Strong: This paper deepens the understanding of some phenomenon or lowers the barriers to an existing research direction.

**Missing References:**

NA

**Paper Topic And Main Contributions:**

The authors proposed a framework named Dual Bank Normalization to address the hubness issue in cross-modal retrieval tasks.
The main contribution can be summarized as follows:
 * The authors find the hubness issue also related to the gallery, which is neglected in existing cross-modal solutions.
 * Two post-processing methods are introduced to normalize the similarities in query bank and gallery bank.
 * The authors give a theoretical demonstration that hubs express high similarity in different data modalities.

**Questions For The Authors:**

Q1: The proposed method shows significant improvements on the VTT task, but I notice the improvements are not significant on the other tasks, i.e., MSVD task. What leads to this difference in the same Text-to-Video setting?

**Reasons To Accept:**

1 Two post-processing methods are proposed to mitigate the hubness by reducing the similarity between hubs and queries.
2 The authors conduct extensive experiments to evaluate their proposed method on eight cross-model retrieval datasets.
3 The paper is well-written and easy to understand.

**Reasons To Reject:**

* The proposed methods (DualIS and DualDIS) are not generic on some cross-model retrieval tasks, i.e., the performance in MSVD (Table 3) shows minor improvements.
* I think the proposed gallery bank is supplementary and less effective compared to the query bank to address hubness issues in cross-model retrieval tasks. This conclusion is also verified by the authors' experiments.

**Reproducibility:**

4: Could mostly reproduce the results, but there may be some variation because of sample variance or minor variations in their interpretation of the protocol or method.

**Reviewer Confidence:**

2: Willing to defend my evaluation, but it is fairly likely that I missed some details, didn't understand some central points, or can't be sure about the novelty of the work.

**Typos Grammar Style And Presentation Improvements:**

Overall, the paper is well writen to read and understand.
* I suggest more details can be provided in algorithm 1 and improve its presentation.
* The 4th contribution (open-source) can be moved to the footnote.
* Highlight RQ1 and RQ2 in bold should be better (pp. 2).

---

> ### Author Rebuttal · Authors · 2023-08-24
>
> We sincerely thank you for your time, efforts, and your detailed and positive comments. We have carefully considered your comments and have provided responses to specific comments.
>
> **Questions For The Authors [1, Performance] / Reasons To Reject [1]:**
>
> > Thank you for your question!
> >
> > 1. Our proposed methods, DualIS and DualDIS, rely on the **normalization of similarities** based on both query and gallery banks. It's important to acknowledge that the **sampling bias of query and gallery banks** could lead to unstable performance under certain scenarios. Additionally, in the design of DualDIS, which normalizes only specific similarities within a subset chosen by a small bank, the sampling bias has been amplified which could lead to suboptimal distribution estimation for embeddings. As suggested by our theorems, a larger bank can lead to better performance because a more precise estimation of distribution can be made.
> >
> >2. On the other side, in our evaluation, we did observe that DualIS and DualDIS, from a **statistical perspective (hubness score)**, perform better than QBNorm (IS and DIS) in mitigating the hubness issue as shown in Table 6 and Table 13. We believe that the retrieval performance is due to the misalignment in retrieval performance and the hubness score, along with the limitation posed by the number of available test data. This is discussed in the Limitations section, where we acknowledge that the hubness score (skewness score) does not exhibit an absolute correlation with retrieval performance (Line 512 to Line 515).
> >
> > We will include this analysis in our final version paper.
>
> **Reasons To Reject [2, Effectiveness of gallery bank]:**
>
> >Thank you for your comments! We agree. The effectiveness of the gallery bank is somewhat overshadowed by the presence of a modality gap, as highlighted in [1]. Modality gap contributes to the challenges in accurately estimating distribution and alignment. But we do believe that a large bank (a more precise estimation) will better reduce the hubness problem.
>
> **Typos Grammar Style And Presentation Improvements:**
>
> >Thank you for your suggestions! We will carefully revise our paper based on your suggestions!
>
> We sincerely thank you for your thoughtful comments and are open to further discussion on these matters.
>
> [1] Weixin Liang, Yuhui Zhang, Yongchan Kwon, Serena Yeung, and James Zou. 2022. Mind the gap: Understanding the modality gap in multi-modal contrastive representation learning. In Advances in neural information processing systems.

---

### Official Review · Reviewer_rioa · 2023-08-05

**Soundness:** 4

**Excitement:**

4: Strong: This paper deepens the understanding of some phenomenon or lowers the barriers to an existing research direction.

**Paper Topic And Main Contributions:**

**Topic:** This paper aims to address the hubness problem in Cross-Modal Retrieval.

**Contribution:**
1. provide the  necessity of incorporating both the gallery and query data;
2. propose a post-processing method that  contains  Dual bank Normalization, Dual Inverted Softmax and Dual Dynamic Inverted Softmax;
3. provide experimental results on multiple task.

**Reasons To Accept:**

1. theoretical analysis of the hubness problem in Cross-Modal Retrieval.
2. the experimental results on multiple task.
3. the post-processing method that integrates multiple improvements.

**Reasons To Reject:**

1. The paper lacks statistics about the hubness problem in Cross-Modal Retrieval.
2. It lacks the analysis about some lower performance with DBNorm (DualDIS).

**Reproducibility:**

3: Could reproduce the results with some difficulty. The settings of parameters are underspecified or subjectively determined; the training/evaluation data are not widely available.

**Reviewer Confidence:**

4: Quite sure. I tried to check the important points carefully. It's unlikely, though conceivable, that I missed something that should affect my ratings.

---

> ### Author Rebuttal · Authors · 2023-08-24
>
> We sincerely thank you for your time, efforts, and your detailed and positive comments. We have carefully considered your comments and have provided responses to specific comments.
>
> **Reasons To Reject [1, Statistics about hubness in CMR]:**
>
> > Thank you for your suggestion!
> > 1. In the submitted paper, following [1], we included the **skewness score, or hubness score**, as a quantitative measure of the hubness problem. Comprehensive data on these scores across various benchmarks can be found in Table 6 (Page 7) of the main paper and Table 13 (Page 22) in the appendix. Notably, we observed a notable reduction in the skewness score, after employing DualIS and DualDIS, proving the effectiveness of our methods in alleviating hubness.
> > 2. To provide a visual representation of the **distribution of 1-occurrence**, we have included them in Figure 3 (Page 8) in the main paper and Figure 6 (Page 21) in the appendix. Intuitively, these figures indicate a clear pattern where a majority of samples are retrieved only once, while a smaller subset of samples experiences frequent retrievals.
>
> **Reasons To Reject [2, Analysis about some lower performance of DualDIS compared to our proposed DualIS]:**
>
> >Thank you for your suggestion! Due to the limitation of space, we were regrettably unable to present a comprehensive analysis of our proposed methods. We will include a detailed analysis in the final version of our paper.
> >
> >1. As DualDIS only normalizes certain similarities within a subset chosen by a small bank, the sampling bias of that bank could lead to worse performance under certain scenarios compared with DualIS. As suggested by our theorems, a larger bank always leads to better performance because a more precise estimation of distribution can be made.
> >
> >2. On the other side, in our evaluation, we did observe that DualDIS, from a statistical perspective (hubness score), performs nearly as well as DualIS in mitigating the hubness issue. Both sets of results are illustrated in Table 6 and Table 13. We believe that the small performance drop of DualDIS compared with our proposed DualIS is due to the misalignment in retrieval performance and the hubness score. This is discussed in the Limitations section, where we acknowledge that the hubness score (skewness score) does not exhibit an absolute correlation with retrieval performance (Line 512 to Line 515).
>
> We sincerely thank you for your thoughtful comments and are open to further discussion on these matters.
>
> [1] Milos Radovanovic, Alexandros Nanopoulos, and Mirjana Ivanovi. 2010. Hubs in space: Popular nearest neighbors in high-dimensional data. Journal of Machine Learning Research, 11(86):2487–2531.

---

### Meta-Review · Area_Chair_P2Gh · 2023-09-19

**Recommendation:** 4

**Metareview:**

This paper proposes a  new query and gallery banks method to mitigate hubness in cross-modal retrieval. The research is original and the methodology clear. The results and experiments are well-supported.

Pros:
* The theoretical analysis of the Hubness in a Cross-Modal Retrieval scenario
* Extensive experimentation for different Cross-Modal retrieval datasets.

Cons:
* Minor improvements when compared to previous research.

---

### Decision · Program_Chairs · 2023-10-07

**Decision:**

Accept-Main

**Comment:**

This paper proposes a  new query and gallery banks method to mitigate hubness in cross-modal retrieval. The research is original and the methodology clear. The results and experiments are well-supported.

Pros:
* The theoretical analysis of the Hubness in a Cross-Modal Retrieval scenario
* Extensive experimentation for different Cross-Modal retrieval datasets.

Cons:
* Minor improvements when compared to previous research.